# Discovering Generalizable Governing Equations for Graph Dynamical Systems with Interpretable Neural Networks

## Abstract

The discovery of symbolic governing equations is a central goal in science; yet, it remains a formidable challenge, particularly for graph dynamical systems, where the network topology further shapes the system behavior. While artificial intelligence offers powerful tools for modeling these dynamics, the field lacks a rigorous comparative benchmark to assess the true scientific utility of the discovered laws. This work establishes the first rigorous benchmark for this task, moving beyond simple fitting metrics to evaluate discovered laws based on their long-term stability and, critically, their out-of-distribution generalization to unseen graph topologies. We introduce the Graph Kolmogorov-Arnold Network (GKAN-ODE), an architecture tailored for this domain, and propose a structure-aware symbolic regression method to leverage its inherent interpretability. Across a suite of synthetic and real-world graph dynamical systems, we demonstrate that symbolic models extracted from neural architectures, particularly our GKAN-ODE, achieve state-of-the-art performance and generalize to unseen networks, significantly surpassing existing baselines. This work presents the first systematic benchmark in this domain, clarifying the expressivity-interpretability trade-offs and offering a pathway from observational data to fundamental physical understanding, providing a critical new tool for data-driven discovery in network science.

## 1 Introduction

The pursuit of scientific knowledge is undergoing a profound transformation, driven by the confluence of vast datasets and sophisticated computational tools. In this "Fourth Paradigm" of science (Hey et al., 2009), Artificial Intelligence (AI) promises not only to accelerate discovery but also to fundamentally change its nature (Wang et al., 2023). The vision extends beyond creating models with high predictive accuracy; the true frontier lies in developing AI that can help us understand the world, unveiling the underlying principles and causal mechanisms that govern complex phenomena (Camps-Valls et al., 2023). This ambition, however, is often hindered by the "black-box" nature of deep learning models, whose internal workings are largely opaque, creating a barrier between computational power and human understanding (Rudin, 2019).

This is challenging, especially in the study of *graph dynamical systems* (Barrat et al., 2008). These systems, where entities interact with each other according to a network structure, are ubiquitous in science, from gene regulatory networks and neural circuits to the spread of epidemics and social dynamics (Barabási, 2013). While we can often observe their evolution, the fundamental laws governing their behavior frequently remain unknown and are heavily dependent on the specific graph instance. Our central objective is to move beyond mere simulation by discovering the symbolic governing *Ordinary Differential Equations* (ODEs) that dictate their evolution directly from observational data.

*Symbolic Regression* (SR) (Makke & Chawla, 2024) emerges as the natural instrument for this task. While traditional evolutionary algorithms and modern sparsity-based frameworks have laid crucial groundwork, the advent of deep learning has opened new possibilities. *Neural Networks* (NNs), with their ability to approximate arbitrary nonlinear functions, can learn the underlying dynamics with

high fidelity. However, this expressivity typically comes at the cost of interpretability, requiring a separate post-hoc SR step to distill symbolic knowledge from the opaque models (Cranmer, 2023).

Despite these advances, a critical gap persists in the literature. The landscape of neural-based equation discovery for graph dynamics is fragmented, with various approaches proposed but no systematic comparative assessment of their performance under different conditions. Researchers seeking to apply these powerful tools lack a clear reference for which architecture to choose, how to implement it, and how to evaluate the scientific plausibility of the discovered equations. Furthermore, the potential of a novel and interpretable-by-design architecture like *Kolmogorov-Arnold Networks* (KANs) by Liu et al. (2025) remains unexplored in this field, despite their demonstrated potential for scientific discovery in other domains (Liu et al., 2024; Koenig et al., 2024).

This paper aims to fill this gap. We present a rigorous, comparative study designed to unveil the actual performance of neural-based models for equation discovery on graph dynamical systems. Our contributions are fourfold:

1. **We provide a rigorous and reproducible benchmark** of state-of-the-art methods, including a leading sparse regression algorithm and *Multilayer perceptron-based* architectures (MLPs). By making our code and experimental setup publicly available, we establish a firm baseline for future research [1].

2. **We introduce the Graph KAN-ODE (GKAN-ODE)**, a novel adaptation of Kolmogorov-Arnold Networks for graph dynamics. We enhance the standard architecture with hyperparameter-free multiplicative nodes to better capture physical interactions and propose a principled, structure-aware *Spline-Wise* symbolic regression algorithm to distill faithful formulas directly from KAN architectures.

3. **We conduct extensive experiments** on both synthetic systems with known ground truths and challenging real-world epidemic data. Our evaluation hinges on a stringent **long-term trajectory rollout metric**, which assesses the stability of the discovered laws that go beyond simple one-step prediction accuracy. Moreover, we demonstrate that the learned symbolic models **generalize effectively to out-of-distribution settings** in unseen scenarios, highlighting their robustness and scientific plausibility.

4. **We offer a critical analysis of the expressivity-interpretability trade-off**. By comparing the symbolic equations extracted from different architectures, we provide practical observations for researchers, clarifying how model choice impacts the complexity and scientific plausibility of the discovered laws.

This work, therefore, serves as both a methodological contribution and a comprehensive benchmark guide, aiming to equip the scientific community with the tools and insights needed to turn observational data into a fundamental understanding of complex systems.

## 2  RELATED WORKS

### 2.1  SYMBOLIC REGRESSION FOR SCIENTIFIC DISCOVERY

Symbolic regression is a methodology for discovering mathematical expressions from data. Unlike standard regression, which fits parameters to a predefined model, SR searches the space of possible expressions $f_{SYM} \in \mathcal{F}$ to find one that optimally balances predictive accuracy and simplicity. Formally, a SR method takes a dataset of input–output pairs $\{(x, y) \mid y = f(x)\}$ and gives a symbolic approximation of $f$, i.e., $\text{SR} : \{(x, y)\} \mapsto \hat{f}_{SR} \approx f$.

Historically, this field was dominated by evolutionary methods like *Genetic Programming* (GP) (Schmidt & Lipson, 2009; Cranmer, 2023), which, while powerful, often face scalability challenges. A prominent alternative is the *Sparse Identification of Nonlinear Dynamics* (SINDy) framework (Brunton et al., 2016), which recasts equation discovery as a sparse regression problem over a library of candidate functions. For network systems, TPSINDy extends this by modeling the system's dynamics as a two-part sparse regression problem, finding separate expressions for the self-dynamics and interaction components (Gao & Yan, 2022).

---

[1]Anonymized code is available at https://anonymous.4open.science/r/Kan-for-Interpretable-Graph-Dynamics-4499/README.md

## 2.2 Deep Learning for Equation Discovery on Graphs

One of the first attempts to leverage NNs to learn analytical expressions was the development of equation learner (EQL) networks (Martius & Lampert, 2017), in which non-linear activation functions are replaced by primitive functions, analogous to SR. Another remarkable work is *AI-Feynman* (Udrescu et al., 2020), an algorithm that combines SR and NN fitting with a suite of physics-inspired techniques that outperformed previous benchmarks. A pivotal contribution by Cranmer et al. (2020) showed that *Graph Neural Networks* (GNNs) can effectively learn the dynamics of systems of particles, and their learned latent representations can then be distilled into symbolic expressions via post-hoc SR. The recent *Learning Law of Changes* (LLC) framework (Hu et al., 2025) advances this approach for graph dynamical systems. It employs separate MLPs to model the self-dynamics and interaction terms (with an explicit multiplicative bias) and then parses them into symbolic form using a pre-trained transformer. Their results demonstrate significant performance gains over prior SR techniques for network dynamics, establishing a key state-of-the-art contribution. However, these methods rely on standard MLPs, whose opaque nature complicates the extraction of interpretable symbolic forms, necessitating a model-agnostic, post-hoc SR step.

## 2.3 Kolmogorov-Arnold Networks: A Path Towards Interpretability

KANs (Liu et al., 2025) have a fundamentally different architecture than MLPs: they place learnable, univariate activation functions, parameterized as splines $\phi$, on the network's edges, while nodes simply perform summation. This design shifts complexity from matrix multiplications and nonlinear activations to a set of univariate functions that can be individually visualized, analyzed, and symbolically regressed. Further technical details can be found in the original paper or in the Appx. A.1. The potential of KANs for scientific discovery has been demonstrated in learning PDE solutions (Liu et al., 2024) and discovering physical laws in dynamical systems without an explicit interaction structure (Koenig et al., 2024). However, to our knowledge, KANs have not yet been applied to discover the governing equations of graph dynamical systems, where network topology drives the evolution of node states over time. Their use has been limited to other graph-based tasks (Bresson et al., 2025), not to the specific challenge of discovering underlying temporal dynamics.

# 3 Methods

This section details our proposed framework for equation discovery. We first establish the formal context for our work by defining graph dynamical systems. Next, we describe the general neural training pipeline, then introduce our Graph KAN-ODE (GKAN-ODE) architecture, and finally outline the symbolic regression procedures and evaluation protocol.

## 3.1 Mathematical Formulation and Notation

The systems under investigation are graph dynamical systems or dynamical processes on complex networks. Such a system is defined by a graph $\mathcal{G} = (\mathcal{V}, \mathcal{E})$, where $\mathcal{V}$ is a set of $N$ nodes (or components) and $\mathcal{E}$ is a set of edges representing their interactions. The state of each node $i \in \mathcal{V}$ at time $t \in \{0, \ldots, T\}$ is described by a vector $\mathbf{x}_i(t) \in \mathbb{R}^d$, while the whole system state is defined as $\mathbf{X}(t) \in \mathbb{R}^{N \times d}$. The graph topological structure can be represented by the adjacency matrix $A \in \mathbb{R}^{N \times N}$, where each entry denotes the connection strength between nodes $i$ and $j$, and $A_{ij} = 0 \iff e_{ij} \notin \mathcal{E}$. As in related works, we focus on graphs with *static* topology, where $\forall t \ A(t) = A$, and in a time-invariant context in which the temporal dynamics of a node $\mathbf{x}_i(t)$ are described by an autonomous ODE:

$$\frac{d\mathbf{x}_i}{dt} = f\left(\mathbf{x}_i, \{\mathbf{x}_j\}_{j \in \mathcal{N}(i)}\right) = \dot{\mathbf{x}}_i \quad \forall t, \tag{1}$$

where $\mathcal{N}(i)$ denotes the neighborhood of node $i$. For clarity, we will omit the explicit time dependence of $\mathbf{x}_i(t)$ hereafter, unless when denoting data points. Following the principle of universality in network dynamics (Barzel & Barabási, 2013) for pairwise interactions, the governing function $f$ can be decomposed into two fundamental components: an intrinsic *self-dynamics* function $H : \mathbb{R}^d \to \mathbb{R}^d$ and an *interaction* function $G : \mathbb{R}^d \times \mathbb{R}^d \to \mathbb{R}^d$ that aggregates effects from neigh-

boring nodes. The dynamics of any node $i$ can thus be expressed as:

$$\dot{\mathbf{x}}_i = H(\mathbf{x}_i) + \sum_{j=1}^{N} A_{ij}\, G(\mathbf{x}_i, \mathbf{x}_j). \tag{2}$$

The primary objective of this work is to discover the symbolic forms of both $H$ and $G$ from discrete-time observations $\{\mathbf{X}(t)\}_{t=0}^{T}$. Models and estimated quantities are denoted with a hat, e.g., $\hat{H}, \hat{\mathbf{x}}_i$.

## 3.2 Learning Dynamics on Graphs with Neural Models

Our primary data consist of time series of graph states $\{\mathbf{X}(t)\}_{t=0}^{T}$, representing discrete measurements of an underlying continuous process. As a prerequisite for learning, we require an estimate of the instantaneous rate of change, the time derivative $\dot{\mathbf{X}}(t)$. We compute a numerical value of the time derivative for each node $\mathbf{x}_i$ using the five-point stencil method (Gao & Yan, 2022), a choice that balances accuracy with robustness to noise in the observational data. This yields a corresponding sequence of derivative evaluations $\{\dot{\mathbf{X}}(t)\}_{t=0}^{T}$. We then train a neural framework to learn the mapping from the system's state $\mathbf{X}(t)$ to its derivative $\dot{\mathbf{X}}(t)$. Following the decoupled formulation in Eq. 2, we parameterize the self-dynamics $H$ and interaction dynamics $G$ with two distinct neural networks, $\hat{H}$ and $\hat{G}$, respectively. The models are trained via gradient descent to minimize the Mean Absolute Error (MAE) loss function between the numerically estimated derivatives $\dot{\mathbf{X}}(t)$ and the model's predictions $\hat{\dot{\mathbf{X}}}(t)$ over the entire training set.

## 3.3 Graph Kolmogorov-Arnold Networks for ODE Discovery

We propose and assess a novel approach, the GKAN-ODE framework, where functions $\hat{H}$ and $\hat{G}$ are parameterized by distinct KANs. In line with the principle that physical laws are often sparse (Brunton et al., 2016), we include the KAN-specific $L^1$ sparsity penalty (Liu et al. (2025) and Appx. A.1) to encourage both $\hat{H}$ and $\hat{G}$ networks to prune inactive splines.

To better capture the multiplicative relationships common in physical dynamics, we further enhance the standard KAN architecture. While prior work has introduced dedicated multiplication layers *between* KAN layers (Liu et al., 2024), this adds structural hyperparameters, requiring prior knowledge or extensive tuning. To circumvent this, we propose a more integrated extension where multiplication occurs *within* each KAN layer. Specifically, for a KAN layer with $d_{out}$ output neurons, we designate half $\lceil d_{out}/2 \rceil$ as standard additive nodes and the remaining $\lfloor d_{out}/2 \rfloor$ as multiplicative nodes. This design allows the model itself, guided by data and sparsity, to learn the appropriate functional form (additive, multiplicative, or a combination) without additional hyperparameters. Our empirical findings, detailed in Appx. C.3, confirm that sparse training effectively prunes multiplicative nodes when the dynamics are purely additive and retains them when they are essential, leading to improved performance over the original architecture.

## 3.4 Symbolic Regression Procedures

Once a neural model is trained, we extract symbolic formulas using two distinct strategies: a model-agnostic, black-box approach and a structure-aware, white-box approach exclusive to KANs.

### 3.4.1 Black-Box Symbolic Regression

A black-box SR method takes data and a model as input and produces symbolic expressions approximating the model predictions. Notably, this procedure treats the models as opaque functions, making it applicable to any machine learning method. In our case, given the trained neural networks $\hat{H}$ and $\hat{G}$, we first generate input-output pairs by performing a forward pass over the training data: $\{\mathbf{x}_i(t), \hat{H}(\mathbf{x}_i(t))\}$ and $\{(\mathbf{x}_i(t), \mathbf{x}_j(t)), \hat{G}(\mathbf{x}_i(t), \mathbf{x}_j(t))\}$ for all interacting pairs. We then fit a separate SR model to each set to obtain symbolic expressions $\hat{H}_{SR}$ and $\hat{G}_{SR}$:

$$\text{SR}(\{\mathbf{x}, \hat{H}(\mathbf{x})\}) = \hat{H}_{SR}, \quad \text{SR}(\{(\mathbf{x}_i, \mathbf{x}_j), \hat{G}(\mathbf{x}_i, \mathbf{x}_j)\}) = \hat{G}_{SR}. \tag{3}$$

The final symbolic model of the full ODE, $f_{SR} \approx \dot{\mathbf{x}}_i$, is constructed by composing these two discovered expressions according to the governing structure of Eq. 2.

### 3.4.2 SPLINE-WISE SYMBOLIC REGRESSION FOR KANS

The architecture of KANs enables a more granular and transparent approach: instead of regressing on the network's aggregate output, we can distill expressions from its elementary components, i.e., the univariate spline activations $\phi$. To fully leverage the transparent structure of KANs, we propose a novel *Spline-Wise* (SW) symbolic regression algorithm for KAN-based models that systematically converts a trained KAN into a fully symbolic equation. While drawing inspiration from prior work (Liu et al., 2025), our procedure incorporates a principled trade-off between expression complexity and accuracy. The procedure is as follows:

1. **Affine Function Fitting.** Given a trained KAN, let $\mathcal{S}$ be the set of all its spline activations after pruning. For each spline $\phi \in \mathcal{S}$, we test its fit against a library $\mathcal{F}$ of candidate univariate symbolic functions. For each candidate function $f \in \mathcal{F}$, we find the optimal affine transformation parameters $\theta_{f,\phi}^* = (a, b, c, d)$ by non-linear least squares that minimize the squared error between the spline's output and the transformed candidate function $f_\phi(x; \theta) = a \cdot f(b \cdot x + c) + d$ over a training set.

2. **Complexity-Penalized Function Selection.** For each spline, we must now select the best symbolic representation from the fitted candidates. We search for the function $f_\phi(x; \theta_{f,\phi}^*)$ that minimizes a penalized error, balancing approximation accuracy with structural complexity. Specifically, let $\Gamma$ be a range of regularization hyperparameters. For each $\phi \in \mathcal{S}$ and $\gamma \in \Gamma$, we search for the function $f \in \mathcal{F}$ that minimizes:

$$f_{\phi,\gamma}^* = \arg\min_{f \in \mathcal{F}} \left[ \text{MSE}\left(\phi(x), f_\phi(x; \theta_{f,\phi}^*)\right) + \gamma \cdot \text{Complexity}(f, \theta_{f,\phi}^*) \right] \quad (4)$$

where MSE is the Mean Squared Error, and $\text{Complexity}(f, \theta_{f,\phi}^*)$ denotes the structural complexity of $f$, defined as the amount of its operators.

3. **Pareto-Optimal Formula Selection.** The previous step yields a set of $|\Gamma|$ candidate symbolic functions for each spline, representing a Pareto front of accuracy versus complexity. We automatically select the optimal function for each spline $f_\phi^*$ by identifying the expression with the highest performance-complexity score, defined as the negative gradient of the log-MSE with respect to complexity (Cranmer, 2023). Its maximum isolates the point at the Pareto curve where the gain in accuracy for an increase in complexity is the highest.

4. **Symbolic Model Reconstruction.** Finally, we replace each spline $\phi$ in the trained KANs $\hat{H}$ and $\hat{G}$ with its selected symbolic counterpart $f_\phi^*$. By composing these elementary functions according to the KANs' architectures, we reconstruct the complete symbolic formula $f_{SW}$, following the structure of Eq. 2.

The pseudo-code algorithm of the above procedure can be found in Appx. A.4, while in Appx. C.4 we show that this approach achieves a more favorable trade-off between accuracy and formula complexity than the SR method proposed by KAN's authors.

### 3.5 EVALUATION METRIC

The ultimate test of a discovered dynamical law is its ability to forecast the system's evolution. Our primary performance measure is, therefore, the MAE between ground-truth trajectories and the predictions obtained by numerically integrating the learned symbolic dynamics. Formally, given a sequence of observations $\{\mathbf{X}(t)\}_{t=0}^T$, let $\hat{H}_{SR}$ and $\hat{G}_{SR}$ be the extracted symbolic formulas. Since they describe the structure of an ODE, we can integrate them over any time interval $[t_0, t_m] \subseteq [0, T]$:

$$\hat{\mathbf{x}}_i(t_m) = \mathbf{x}_i(t_0) + \int_{t_0}^{t_m} \left[ \hat{H}_{SR}\left(\hat{\mathbf{x}}_i(t)\right) + \sum_{j=1}^N A_{ij} \hat{G}_{SR}\left(\hat{\mathbf{x}}_i(t), \hat{\mathbf{x}}_j(t)\right) \right] dt. \quad (5)$$

Our assessment begins with a given set of initial conditions $\mathbf{X}(t_0)$ from a test trajectory, which are then used to integrate the symbolic model via Eq. 5 for all subsequent time steps, resulting in a predicted trajectory $\{\hat{\mathbf{X}}(t)\}_{t=t_0+1}^{t_m}$. We then compute the trajectory mean absolute error, $\text{MAE}_{\text{traj}}$, between the ground-truth observations and predictions:

$$\text{MAE}_{\text{traj}} = \frac{\sum_{i=1}^N \sum_{t=t_0}^{t_m} |\mathbf{x}_i(t) - \hat{\mathbf{x}}_i(t)|}{N(t_m - t_0 - 1)}. \quad (6)$$

This integration is autoregressive, meaning that prediction errors at one step are propagated into the next. Consequently, even minor inaccuracies in the discovered equations can compound over time, making the $\text{MAE}_{\text{traj}}$ a stringent and comprehensive test of a model's long-term accuracy and stability. Furthermore, this metric does not rely on prior knowledge of the true governing equations, thereby making it more suitable for real-world scenarios.

# 4 EXPERIMENTAL DESIGN

This section outlines the empirical framework for assessing equation discovery methods in graph dynamical systems, detailing the models, datasets, and evaluation metrics for performance, symbolic accuracy, and generalization. The Appendices and source code offer further information on dataset generation, model implementation, optimization, SR algorithms, and hyperparameters, ensuring scientific reproducibility and fairness.

## 4.1 MODELS UNDER ASSESSMENT

We rigorously and fairly assess a set of distinct state-of-the-art methodologies for inferring the governing equations of dynamical systems on graphs. In addition to the proposed GKAN-ODE model, we test three other approaches: our own implementation of a MLP-ODE, a GMLP-ODE model, the neural architecture of LLC, and the TPSINDy algorithm. The MLP-ODE is a simple baseline that models node dynamics $\dot{\mathbf{x}}_i$ relying solely on the local state $\mathbf{x}_i$, effectively ignoring neighbor interactions and quantifying the specific contribution of topological information to the discovery process. The GMLP-ODE serves as the direct MLP-based counterpart of GKAN-ODE, where the two KANs are replaced by MLPs, allowing for a controlled comparison between the two architectures. LLC is included as a state-of-the-art neural baseline, notable for its MLP-based architecture that explicitly introduces multiplication in the network's structure for $\hat{G}$ in a manner conceptually similar to GKAN-ODE. Unlike these neural approaches, TPSINDy directly learns sparse symbolic expressions for $\hat{H}$ and $\hat{G}$ from data and represents the leading non-neural approach. For the neural architectures, we utilize SR procedures to extract interpretable equations. As a black-box SR, the GP-based tool PySR (Cranmer, 2023) is employed, and the resulting symbolic models are labeled with the suffix "+GP". Similarly, the SW fitting applied to our proposed model is referred to as GKAN-ODE+SW.

## 4.2 INFERENCE ON SYNTHETIC DYNAMICAL SYSTEMS

We first evaluate the models' capacity to recover the precise symbolic form of known dynamics. To this end, we utilize four canonical network dynamical systems, chosen to represent a diverse range of nonlinearities common in scientific models (Barzon et al., 2024): Kuramoto oscillators (KUR), epidemic spreading (EPID), biochemical (BIO), and population (POP) dynamics. We generate these synthetic datasets by integrating the models on a fixed Barabási–Albert (Barabási & Albert, 1999) network, chosen for its scale-free topology, which is representative of many real-world systems. To evaluate robustness against measurement uncertainty, we also create noisy variants of these systems by adding white noise to node states at each time step under different signal-to-noise ratio (SNR) levels. For all experiments with this setting, models are trained on the first 80% of the temporal observations, with the remaining 20% reserved for validation and hyperparameter tuning.

A crucial component of our methodology is the rigorous selection of the final symbolic model. Recognizing that models may overfit to a specific network instance, and that both the GP-based and the SW fitting procedures are sensitive to hyperparameters, we design a robust validation framework. We generate an additional validation set by simulating the same dynamics on a *new* graph with a different topology and initial conditions. For each candidate symbolic formula produced by the SR algorithms, we compute the trajectory rollout error (Eq. 6) on this out-of-distribution (OOD) *validation set*. The symbolic form achieving the lowest $\text{MAE}_{\text{traj}}$ is selected as the definitive expression representing the underlying ODE. Extending the related works, we aim to assess the generalization of trained models and extracted equations in a novel context: a final *test set* that includes three unique simulations, each with distinct graph topologies and random initial conditions. We report the $\text{MAE}_{\text{traj}}$ averaged over these three test trajectories for both models and formulas, indicating

their generalization beyond the training domain. A detailed visualization of the entire experimental pipeline is provided in Appx. B.2.

### 4.3 Inference on Real-World Empirical Data

To assess performance on a task with unknown ground truth, we utilize the empirical dataset of epidemic dynamics from Gao & Yan, which captures the early pre-intervention spread of the H1N1, SARS, and COVID-19 outbreaks across the global airline network. We train the neural models on the COVID-19 dataset and extract symbolic representations. As a true OOD validation set is unavailable, we select the symbolic expression that yields the lowest $\text{MAE}_{\text{traj}}$ on the training data itself. This procedure discovers a single homogeneous equation describing the global average dynamics. To account for country-specific variations, we then fine-tune the coefficients of this discovered symbolic structure for each node, following the ideas proposed in prior works (Gao & Yan, 2022; Hu et al., 2025) and detailed in Appx. B.3. Our evaluation focuses on the generalizability of the discovered laws. We investigate whether the symbolic structures learned from COVID-19, with only coefficient fine-tuning, can effectively model H1N1 and SARS outbreaks. For final model assessment, we utilize the long-term trajectory rollout metric, $\text{MAE}_{\text{traj}}$, and compare it with previous studies using a short-term, single-step prediction metric, $\text{MAE}_{\text{eul}}$, which relies on the Euler method with ground-truth data rather than prior model predictions, emphasizing short-term accuracy and reducing the impacts of long-term instability.

## 5 Results and Discussion

### 5.1 Comparative Performance on Synthetic Systems

Our first key finding, illustrated in Fig. 1 (left), is the superior performance of neural-based architectures over the sparse regression baseline, TPSINDy, and topology-agnostic baselines. MLP-ODE suffers from catastrophic error accumulation across most dynamics, demonstrating that the governing laws are inextricably linked to the specific network topology and cannot be resolved by simple curve-fitting or mean-field approximations. The graph-aware neural models, both before and after symbolic distillation, consistently yield more accurate and stable long-term trajectory rollouts, as measured by the $\text{MAE}_{\text{traj}}$. TPSINDy correctly identifies the KUR dynamics, arguably due to its expressiveness in periodic functions; however, it fails in the other cases. Notably, on the EPID and POP datasets, the naive MLP-ODE yields lower rollout errors than TPSINDy. This counter-intuitive result highlights a critical limitation of restricted sparse regression: while TPSINDy is not compositional and fails by identifying incorrect interaction terms that lead to diverging trajectories, the MLP-ODE learns an approximated function that remains numerically stable. Conversely, neural-based approaches combined with SR can overcome these limitations by composing complex, nested equations starting from a restricted library of univariate functions and simple binary operators. Among the neural approaches, the models derived from our GKAN-ODE architecture demonstrate remarkable efficacy, and its black-box (GKAN-ODE+GP) symbolic model is consistently among the top performers, achieving the lowest rollout error. While the LLC architecture also performs well, particularly compared to its GMLP-ODE counterpart, the GKAN-based models frequently exhibit lower mean error and smaller variance across all time steps (Appx. C.1). Beyond raw performance, GKAN-ODE models are also more parameter-efficient than the baselines: Fig. 1 (right) provides a clear visualization of the trade-off between performance ($\text{MAE}_{\text{traj}}$) and the number of parameters. The figure promotes GKAN-ODE once more as the most promising choice for efficient equation discovery in graph dynamical systems.

### 5.2 Symbolic Discovery and Interpretability

As shown in Tab. 1, the black-box GKAN-ODE+GP procedure demonstrates exceptional capability in recovering the ground-truth dynamics, successfully extracting (up to algebraic transformations) the exact symbolic form of the governing equations for all four synthetic systems. The discovered structures are identical to the ground truth, and the fitted coefficients are remarkably precise, validating the entire pipeline from neural training to symbolic distillation. The structure-aware GKAN-ODE+SW approach offers a more direct window into the model's inner workings. Detailed in Tab. 2, this method also successfully identifies the correct underlying dynamics. For the BIO system, it re-

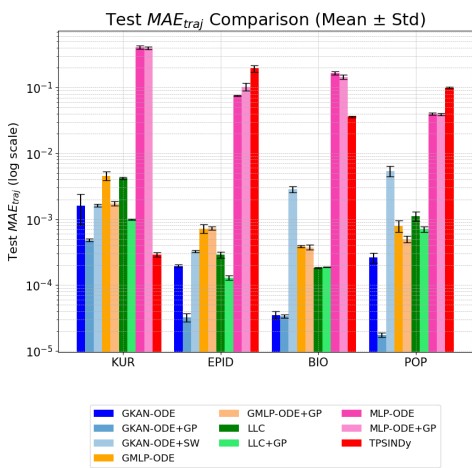 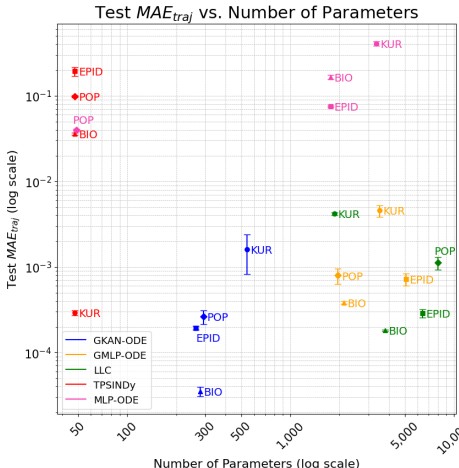

Figure 1: Performance comparison on synthetic dynamics. (Left) Comparison of test MAE$_{\text{traj}}$ for both models and the inferred equations. (Right) Test MAE$_{\text{traj}}$ and number of parameters of the trained neural-based models and TPSINDy (whose parameters are defined by its symbolic function library). Values are averaged on three test graphs and the standard deviation is reported as errors bars.

trieves slightly different coefficients, while for the KUR system, it discovers a phase-shifted sine function that is mathematically equivalent to the ground truth. For more complex dynamics like EPID or POP, the SW method sometimes yields expressions with additional small-coefficient terms or minor parameter deviations (e.g., $< 1\%$). While numerically small, these deviations can accumulate during autoregressive integration, leading to a higher $MAE_{traj}$. However, in the context of scientific discovery, this result remains a success: the method effectively isolates the correct governing structure. The precise calibration of these coefficients can subsequently be handled by standard parameter estimation techniques, making the SW output a highly actionable starting point for physical modeling. While the GP-based approach acts as a surrogate model seeking the simplest approximation of the network's output, the SW method offers a faithful, white-box translation of the network's internal logic. This provides a stronger form of interpretability than standard attribution scores: it establishes a direct and global mapping between input features and dynamical behavior (i.e., explicitly identifying how a neighbor's state $x_j$ drives dynamics). As shown in Appx.C.4, our proposed SW algorithm not only recovers valid symbolic forms but also achieves a significantly better accuracy-complexity trade-off than the original KAN symbolic regression method, effectively filtering out redundant complexity while retaining the model's structural insights. The equations extracted by TPSINDy, GMLP-ODE+GP, and LLC+GP, along with their discussion, can be found in Appx. C.2.

When dealing with dynamics that involve observation noise, neural models are still able to recover competitive expressions under low noise conditions (SNR = 70 dB). However, at higher noise levels (SNR $\leq$ 50 dB), all models tend to degenerate. More details are available in Appx. C.5.

Table 1: Ground-truth and discovered symbolic equations $f_{SR}$ for the four synthetic dynamical systems learned by the best-validated GKAN-ODE+GP model and rounded to four decimal places. The structural complexity matches between ground-truth and learned models: it is 5 for KUR and POP, and 6 for EPID and BIO.

| Dataset | Ground-Truth Equation | GKAN-ODE+GP Discovered Symbolic Expression |
|---|---|---|
| KUR | $2 + \frac{1}{2}\sum_j A_{ij}\sin(x_j - x_i)$ | $1.9992 + \sum_j A_{ij}(-0.5005 \cdot \sin(x_i - x_j))$ |
| EPID | $-\frac{1}{2}x_i + \frac{1}{2}\sum_j A_{ij}(1 - x_i)x_j$ | $-0.4997 \cdot x_i + \sum_j A_{ij}(x_j \cdot (0.5001 - 0.5002 \cdot x_i))$ |
| BIO | $1 - \frac{1}{2}x_i - \frac{1}{2}\sum_j A_{ij}x_i x_j$ | $-0.5006 \cdot x_i + 1.0002 + \sum_j A_{ij}(-0.4998 \cdot x_i x_j)$ |
| POP | $-\frac{1}{2}x_i + \sum_j A_{ij}\frac{x_j^3}{5}$ | $-0.4999 \cdot x_i + \sum_j A_{ij}(0.2000 \cdot x_j^3)$ |

Table 2: Best-validated Spline-wise symbolic formulas $f_{SW}$ and their structural complexity for the GKAN-ODE+SW model on the four synthetic dynamical systems.

| Dataset | GKAN-ODE+SW Discovered Symbolic Expressions | Complexity |
|---|---|---|
| KUR | $1.9991 + \sum_j A_{ij}(-0.5005 \cdot sin(-0.9992 \cdot x_i + 0.9995 \cdot x_j + 3.1373))$ | 8 |
| EPID | $-0.4988 \cdot x_i + \sum_j A_{ij}(-0.4961 \cdot x_i x_j + 0.4970 \cdot x_j - 0.0022 \cdot x_i + 0.0018)$ | 10 |
| BIO | $-0.5000 \cdot x_i + 1.0001 + \sum_j A_{ij}(-0.4899 \cdot x_i x_j)$ | 6 |
| POP | $-0.2862 x_i - 0.1744 \tanh(1.4270 x_i - 0.0779) - 0.0122$ $+ \sum_j A_{ij}(0.1474 x_j^3 + 0.0066 x_j^2 + 0.0204 x_j)$ | 16 |

## 5.3 DISCOVERY IN REAL-WORLD EPIDEMIC DYNAMICS

In this scenario, the symbolic equations derived for the global average dynamics are reported in Appx. D.1. The models yield diverse functional forms, with TPSINDy favoring a logistic-like interaction, while neural architectures learn more complex nonlinearities. This scenario further highlights the critical trade-off between model expressivity and interpretability. Notably, the GKAN-ODE+GP model distills a particularly simple and plausible law, suggesting a linear self-term with an exponential growth interaction from neighbors. In contrast, the GKAN-ODE+SW method produces a significantly more complex but fully transparent expression by directly translating the KAN's internal splines. This presents a choice for domain experts: pursuing the simplest explanatory model (via GP) or analyzing a more faithful, albeit complex, representation of the neural-learned dynamics (via SW). Both are valid pathways to scientific discovery, serving different analytical goals.

Key evaluation rests on the models' stability over time and their generalization to unseen data. Fig. 2 contrasts the performance of the discovered equations' on the COVID-19 test set against their adaptability to H1N1 and SARS dynamics after tuning the coefficients. While TPSINDy is competitive in single-step forecasting (MAE$_{\text{eul}}$), it leads to catastrophic error accumulation in long-term trajectory rollouts (MAE$_{\text{traj}}$). Conversely, all neural-derived laws, and in particular GKAN-ODE models, exhibit strong long-term stability and generalization, which are crucial for identifying scientific models in complex systems.

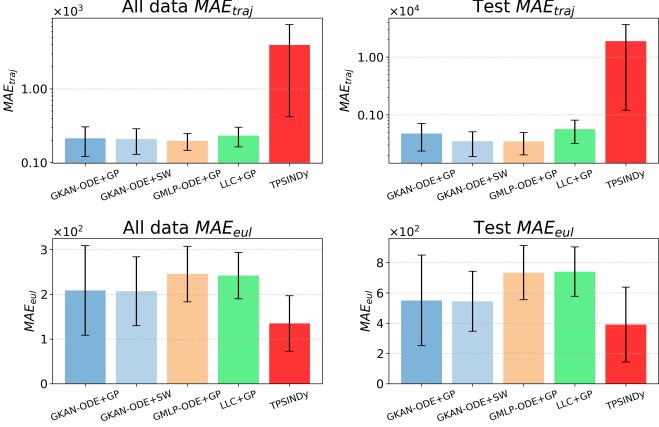

Figure 2: Performance comparison of the symbolic formulas of Tab. 15 averaged on the COVID, H1N1, SARS datasets. Both MAE$_{\text{traj}}$ (top) and MAE$_{\text{eul}}$ (bottom) are computed on the complete (left) and test (right) datasets.

## 6 CONCLUSION

This paper rigorously assesses the most prominent AI methods of equation discovery for graph dynamical systems to reveal their true performance. Our findings establish that our proposed GKAN-ODE models significantly outperform the sparse regression and MLP-based baselines. KAN-based

models further demonstrate a superior balance of predictive accuracy, parameter efficiency, and an architecture inherently amenable to interpretation.

A key contribution of this work lies in the distillation of symbolic knowledge from these models. We have shown how a model-agnostic symbolic regression can effectively recover the ground-truth equations. In parallel, our novel Spline-Wise fitting algorithm provides a transparent and truthful, albeit more granular, symbolic representation of KANs' internal logic. This presents a valuable choice for researchers: a pragmatic path to the most parsimonious symbolic law or a more detailed, faithful representation of what the model has actually learned from the data.

By establishing a reproducible benchmark and advocating for evaluation based on long-term, out-of-distribution generalization, this work aims to serve as a practical reference and open-source contribution for the interdisciplinary scientific community working on complex systems. It clarifies the state-of-the-art and promotes a human-in-the-loop paradigm where AI acts as a powerful collaborator that generates plausible, testable hypotheses, thereby augmenting human intuition and understanding. Future research should focus on adapting such models for time-dependent systems with evolving topology, as well as developing tools for dealing with noisy and more complex real-world systems. Ultimately, this study confirms the viability of interpretable neural architectures as powerful tools for the scientific community in the quest to understand the fundamental laws governing complex systems.

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

APPENDIX

# A  METHODOLOGICAL DETAILS

## A.1  KOLMOGOROV-ARNOLD NETWORK (KAN) BACKGROUND

Kolmogorov-Arnold Networks (KANs), proposed by Liu et al. (2025), are a specific type of neural network that has been recently proposed as a valid alternative to Multi-Layer Perceptrons (MLPs). Whereas MLPs are inspired by the universal approximation theorem, KANs are inspired by the Kolmogorov-Arnold representation theorem (Kolmogorov, 1961; Braun & Griebel, 2009), which states that if $f : [0,1]^d \to \mathbb{R}$ is a multivariate continuous function on a bounded domain, then it can be written as:

$$f(\mathbf{x}) = f(x_1, x_2, ..., x_d) = \sum_{q=1}^{2d+1} \Phi_q \left( \sum_{p=1}^{d} \phi_{q,p}(x_p) \right), \tag{7}$$

where $\phi_{q,p} : [0,1] \to \mathbb{R}$, $\Phi_q : \mathbb{R} \to \mathbb{R}$. In other words, $f$ can be reduced to a suitably defined composition of univariate functions, where the composition only involves simple addition. The underlying idea of KANs is to substitute the weights and fixed activation functions of MLPs with learnable univariate activation functions on edges and sum aggregation on nodes.

The general definition of a KAN layer $\boldsymbol{\Phi}_l$ with $d_{in}$-dimensional input and $d_{out}$-dimensional output consists of a matrix of univariate functions:

$$\boldsymbol{\Phi}_l = \begin{bmatrix} \phi_{l,1,1}(\cdot) & \phi_{l,1,2}(\cdot) & \cdots & \phi_{l,1,d_{in}}(\cdot) \\ \phi_{l,2,1}(\cdot) & \phi_{l,2,2}(\cdot) & \cdots & \phi_{l,2,d_{in}}(\cdot) \\ \vdots & \vdots & \ddots & \vdots \\ \phi_{l,d_{out},1}(\cdot) & \phi_{l,d_{out},2}(\cdot) & \cdots & \phi_{l,d_{out},d_{in}}(\cdot) \end{bmatrix} \tag{8}$$

where $\phi_{l,j,i}$ represents the learnable activation function applied to the $i^{th}$-feature of the input of the $j^{th}$-neuron at layer $l$. After computing all the $d_{in} \cdot d_{out}$ activation values, the output of the $l^{th}$ layer $\mathbf{x}_l \in \mathbb{R}^{d_{out}}$ is obtained by summing along the first dimension of the matrix described in Eq. 8. Stacking multiple KAN layers results in an architecture with a shape represented by an integer array:

$$[d_0, d_1, ..., d_L],$$

where $d_l$ represents the number of neurons in the $l^{th}$-layer and $d_0 = |\mathbf{x}|$. Each univariate function in Eq. 8 has trainable parameters that can be learned through backpropagation and gradient descent. Specifically, they are defined as splines with residual activations.

KANs are usually trained with a sparsity loss, which is an adaptation of the L1 norm of MLPs. However, this norm is directly defined on the learned activation functions. Formally, the L1 norm of an activation function $\phi$ is given by the average magnitude over its $N_p$ inputs, that is:

$$|\phi|_1 \equiv \frac{1}{N_p} \sum_{s=1}^{N_p} |\phi(x^{(s)})|. \tag{9}$$

Then, the L1 norm of a KAN layer $\boldsymbol{\Phi}$ with $d_{in}$ inputs and $d_{out}$ outputs is defined as:

$$|\boldsymbol{\Phi}|_1 \equiv \sum_{i=1}^{d_{in}} \sum_{j=1}^{d_{out}} |\phi_{i,j}|_1, \tag{10}$$

that is, the sum of L1 norms of all the activation functions in $\boldsymbol{\Phi}$. Furthermore, an entropy term is added to the loss definition:

$$S(\boldsymbol{\Phi}) \equiv -\sum_{i=1}^{d_{in}} \sum_{j=1}^{d_{out}} \frac{|\phi_{i,j}|_1}{|\boldsymbol{\Phi}|_1} log \left( \frac{|\phi_{i,j}|_1}{|\boldsymbol{\Phi}|_1} \right). \tag{11}$$

Then, the final training loss $\mathcal{L}_{total}$ is given by the prediction loss $\mathcal{L}_{pred}$ plus the L1 and entropy regularization aggregated over all the layers:

$$\mathcal{L}_{total} = \mathcal{L}_{pred} + \lambda \left( \mu_1 \sum_{l=1}^{L} |\boldsymbol{\Phi}_l|_1 + \mu_2 \sum_{l=1}^{L} S(\boldsymbol{\Phi}_l) \right), \tag{12}$$

Table 3: Hyperparameter ranges of the MLPs in the GMLP-ODE models

| Hyperparameter | Values |
|---|---|
| Hidden dimensions | $[8, 64]$ |
| Activation function | $\{\texttt{relu}, \texttt{softplus}, \texttt{tanh}\}$ |
| Dropout probability | $[0.0001, 0.5]$ |
| Hidden layers | $\{1, 2\}$ |
| Learning rate | $[0.0005, 0.05]$ |
| Batch size | $\{16, 32, 64\}$ |

where $\mu_1, \mu_2$, and $\lambda$ are hyper-parameters that determine the impact of the corresponding loss terms.

One of the key characteristics of KANs is that they can be used to perform symbolic regression. Specifically, once the model is trained, it is possible to prune inactive neurons by looking at their spline activation magnitudes and then fixing the remaining activation functions to symbolic formulas (e.g., $\texttt{sin}$, $\texttt{cos}$) so that the whole model can be described through a symbolic representation. This process enhances interpretability, as it overcomes the black-box nature typical of deep-learning models by providing, as output, a human-readable mathematical formulation of the learned function. Refer to the original paper for further details on the pruning and regression procedures.

## A.2 TECHNICAL IMPLEMENTATION

We implemented the model under consideration using Python 3.12.0, Pytorch 2.3.1, and PyG 2.3.1. For hyper-parameter tuning, we employed the Optuna package (Version 4.3.0). The Spline-Wise fitting procedure relies on the $\texttt{curve\_fit}$ method from the $\texttt{scipy}$ library for solving the non-linear least squares problem, while for the GP-based SR algorithm, we used PySR 1.5.5. We utilized the $\texttt{dopri5}$ solver from the $\texttt{torchdiffeq}$ library as a numerical integrator for computing the rollout metric $\text{MAE}_{\text{traj}}$, setting $\texttt{atol} = \texttt{rtol} = 10^{-5}$ for all models.

## A.3 HARDWARE SETUP

We carried out the experiments on a Google Cloud $\texttt{g2-standard-48}$ virtual machine, equipped with 48 vCPUs based on the Intel Cascade Lake CPU architecture and 192 GB of system memory. The setup was further accelerated by 4 NVIDIA L4 GPUs.

## A.4 SPLINE-WISE SYMBOLIC REGRESSION ALGORITHM

Algorithm 1 describes the proposed Spline-Wise symbolic fitting procedure of KAN-based models. To ensure parsimony, in line 10, the coefficients with magnitudes below a threshold ($\epsilon$) are pruned before complexity is computed. For example, the expression $x^3 + 10^{-5}x^2$ is considered to have a complexity of 1, not 4. As a measure of complexity, we use the $\texttt{count\_ops}$ function from $\texttt{sympy}$ library, which measures the number of operations an expression contains.

## A.5 HYPERPARAMETERS SPECIFICATIONS

This section lists the search spaces of the employed hyperparameters in the experimental analysis. Model selection is performed using the Optuna package, optimizing the MAE over 35 trials for synthetic dynamics, over 70 trials for dynamics with noise, and over 100 trials for COVID-19 data. All neural architectures are optimized using Adam for 1000 epochs with early stopping and patience parameters of 200 for synthetic dynamics and 300 for the real-world COVID dataset. Tabs. 3, 4 and 5 detail the hyperparameter ranges used for the neural-based models. For TPSINDy, we consider the default libraries of symbolic functions provided by the authors in the original implementation, including polynomial, trigonometric, fractional, and exponential terms.

Model selection of GP-based and Spline-Wise SR methods is performed via grid search according to the hyperparameter grids specified in Tabs. 6, 7, respectively.

Table 4: Hyperparameter ranges of the MLPs in the LLC models, where $\lambda$ denotes the regularization parameter of the penalized loss function minimized during training.

| Hyperparameter | Values |
|---|---|
| Hidden dimensions | $[8, 64]$ |
| Activation function | $\{\texttt{relu}, \texttt{softplus}, \texttt{tanh}\}$ |
| Hidden layers | $\{1, 2\}$ |
| Learning rate | $[0.0005, 0.05]$ |
| Batch size | $\{16, 32, 64\}$ |
| Regularization $\lambda$ | $[0.0, 0.01]$ |

Table 5: Hyperparameter ranges of the KANs in the GKAN-ODE models.

| Hyperparameter | Values |
|---|---|
| Grid size | $[5, 20]$ |
| Spline order | $[1, 3]$ |
| Range limit | $[-10, 10]$ |
| Hidden dimensions | $[1, 6]$ |
| Regularization $\lambda$ | $[10^{-6}, 1.0]$ |
| Learning rate | $[0.0005, 0.05]$ |
| $\mu_1$ | $[0.1, 1.0]$ |
| $\mu_2$ | $[0.1, 1.0]$ |
| Batch size | $\{16, 32, 64\}$ |

Table 6: Hyperparameter grid for the PySR algorithm.

| Hyperparameter | Values |
|---|---|
| Number of iterations | $\{50, 100, 200\}$ |
| Model selection | $\{\text{Score}, \text{Accuracy}\}$ |
| Binary operators | $[+, -, *, /]$ |
| Symbolic library $\mathcal{F}$ | $[\texttt{exp, sin, neg, square, cube, abs,}$ $\texttt{tan, tanh, ln, zero}]$ |

Table 7: Hyperparameter grid of the Spline-wise fitting algorithm. The *Model selection* parameter is used at line 28 of Algorithm 1, and defines whether to choose the function with the highest score (thus favoring simpler equations) or with the lowest log loss (thus favoring accuracy).

| Hyperparameter | Values |
|---|---|
| Spline pruning threshold $\rho$ | $\{0.01, 0.05, 0.1\}$ |
| Coefficient pruning threshold $\epsilon$ | $\{0.001, 0.01, 0.1\}$ |
| Model selection | $\{\text{Score}, \text{Log loss}\}$ |
| $\Gamma$ | $[10^{-5}, 10^{-4}, 10^{-2}, 10^{-1}, 1]$ |
| Symbolic library $\mathcal{F}$ | $[\texttt{identity, square, cube, exp, abs,}$ $\texttt{sin, cos, tan, tanh, ln, zero}]$ |

---

**Algorithm 1** Spline-wise Symbolic Regression for KAN

---

**Require:** Splines $\mathcal{S}$, function library $\mathcal{F}$, regularization grid $\Gamma$, training data $(x, y)$, coefficient pruning threshold $\varepsilon$, spline pruning threshold $\rho$
**Ensure:** Selected symbolic function $f_\phi^*$ for each $\phi \in \mathcal{S}$ and final symbolic formula $f_{SW}$
1: $\mathcal{S}_{\mathrm{pruned}} = \mathrm{pruning}(\mathcal{S}, \rho)$
2: **for** each spline $\phi \in \mathcal{S}_{\mathrm{pruned}}$ **do**
3:     Collect inputs $X_\phi$ and outputs $Y_\phi$ from training data
4:     $\mathrm{Results}_\phi = [\cdot]$
5:     **for** each $\gamma \in \Gamma$ **do**
6:         best_state = $(\cdot)$
7:         best_L = $\infty$
8:         **for** each candidate function $f \in \mathcal{F}$ **do**
9:             Fit affine parameters $\theta_{f,\phi}^* = (a, b, c, d)$
10:            Prune negligible coefficients $< \varepsilon$ from $f$
11:            Compute predictions $\hat{Y}_{f,\phi} = f_\phi(X_\phi; \theta_{f,\phi}^*)$
12:            $\mathrm{MSE} = \frac{1}{|X_\phi|} \sum (Y_\phi - \hat{Y}_{f,\phi})^2$
13:            $c = \mathrm{Complexity}(f, \theta_{f,\phi}^*)$
14:            $L = \mathrm{MSE} + \gamma \cdot c$
15:            $\ell = \log(\mathrm{MSE})$
16:            **if** $L < \mathrm{best\_L}$ **then**
17:                best_state = $(f, \theta_{f,\phi}^*, c, \ell)$
18:            **end if**
19:         **end for**
20:         Append best_state to $\mathrm{Results}_\phi$
21:     **end for**
22:     Sort $\mathrm{Results}_\phi$ by complexity $c$
23:     scores = [ $\mathrm{Results}_\phi[0][f]$, 0 ]
24:     **for** each consecutive pair $(c_1, \ell_1), (c_2, \ell_2)$ **do**
25:         $\Delta = (\ell_2 - \ell_1)/(c_2 - c_1)$
26:         Append $-\Delta$ to scores
27:     **end for**
28:     Select $f_\phi^* = $ function with highest score
29: **end for**
30: Combine all $f_\phi^*$ according to KAN's additive/multiplicative structure
31: Build final symbolic formula $f_{SW}$
32: **return** $\{f_\phi^*\}, f_{SW}$

---

# B   Experimental Setup and Datasets

## B.1   Synthetic Dataset Generation and Reproducibility Protocols

Tab. 8 shows the general equations of the studied synthetic dynamical systems, while Tab. 1 reports the considered instantiations. Preliminary experiments with different dynamic parameters—provided they are physically consistent—did not yield significant differences in terms of the models' learning capabilities; hence, we set their magnitude to plausible values considered in the literature (Barzon et al., 2024; Gao & Yan, 2022). The datasets are generated by numerically integrating these models with the Runge–Kutta method of order 5, implemented in the `solve_ivp` function from the `scipy` library, using absolute and relative tolerances of $10^{-12}$ to ensure high numerical precision. We simulate the dynamics on a Barabási–Albert (Barabási & Albert, 1999) graph with 70 nodes and an attachment parameter $m = 3$, saving the solutions at $T = 2000$ regularly spaced time steps. We report in Tab. 9 the set of parameters to reproduce dataset generation.

Table 8: General equations of the considered dynamical processes.

| Dynamics | Equation $dx_i/dt$ |
|---|---|
| KUR | $\omega + K \sum_j A_{ij} \sin(x_j - x_i)$ |
| EPID | $-\mu x_i + \beta \sum_j A_{ij}(1 - x_i)x_j$ |
| BIO | $\alpha - \delta x_i - \kappa \sum_j A_{ij} x_i x_j$ |
| POP | $-r x_i^b + \sigma \sum_j A_{ij} x_j^a$ |

Table 9: Parameters to reproduce the creation of the synthetic datasets.

| Dynamics | Initial condition | $T_{\text{Start}}$ | $T_{\text{End}}$ |
|---|---|---|---|
| KUR | Uniform $[0, 2\pi]$ | 0 | 1 |
| EPID | Uniform $[0, 1]$ | 0 | 2 |
| BIO | Uniform $[0, 1]$ | 0 | 1 |
| POP | Uniform $[-1, 1]$ | 0 | 10 |

For the KUR dynamics, oscillator phases are initialized uniformly in $[0, 2\pi]$ to cover the full angular domain. For EPID and BIO dynamics, node states are chosen in $[0, 1]$ to represent normalized concentrations or infection probabilities. For POP, instead, we chose to initialize nodes in the interval $[-1, 1]$ to better expose the effect of the polynomial term $x^a$, as including both positive and negative values yields richer trajectories. Regarding the temporal horizons $T_{\text{Start}}$ $T_{\text{End}}$, we set them to allow each dynamic to display its full evolution.

The intermediate validation set used to tune the hyper-parameters of the SR algorithms is obtained by simulating the dynamics on an additional Barabási–Albert graph with 100 nodes and the same attachment parameter used for the training graphs. The graphs used to generate the three OOD test sets are a BA graph with 70 nodes (analogous to the one used for training but initialized with a different random seed), a Watts–Strogatz small-world graph with 50 nodes, and an Erdős–Rényi random graph with 100 nodes and an edge probability of 0.05. For validation and test datasets, we simulate the dynamics for $T = 1000$ steps.

All data-generating code, along with its random seeds, is provided in the codebase.

### B.2 EXPERIMENTAL PIPELINE OVERVIEW

To provide a clear visual guide to our evaluation framework, Fig. 3 illustrates the overall process for training, symbolic distillation, and evaluation of all models. This pipeline is designed to assess not only the accuracy of the discovered equations but, more importantly, their long-term stability and generalization capabilities on unseen data, which are critical for establishing scientific utility.

### B.3 REAL-WORLD DATASET AND PREPROCESSING

The considered empirical datasets are based on the epidemiological spread of infectious diseases (SARS, COVID, H1N1), modeled by the worldwide airline network of human mobility between different countries, where each entry of the weighted adjacency matrix represents the traffic volume across regions. Refer to Gao & Yan (2022) for additional details. Regarding the COVID dataset, we normalize the values to the range $(-1, 1)$ using a `MinMax` scaler. The scaler is fitted only on the training set (the first 80% of the data) to prevent data leakage. We perform the same pre-processing steps for the H1N1 and SARS datasets before fine-tuning the coefficients of the learned symbolic formulas using neural models. During the evaluation phase, the same scaler is applied to transform the predicted values back to the original scale.

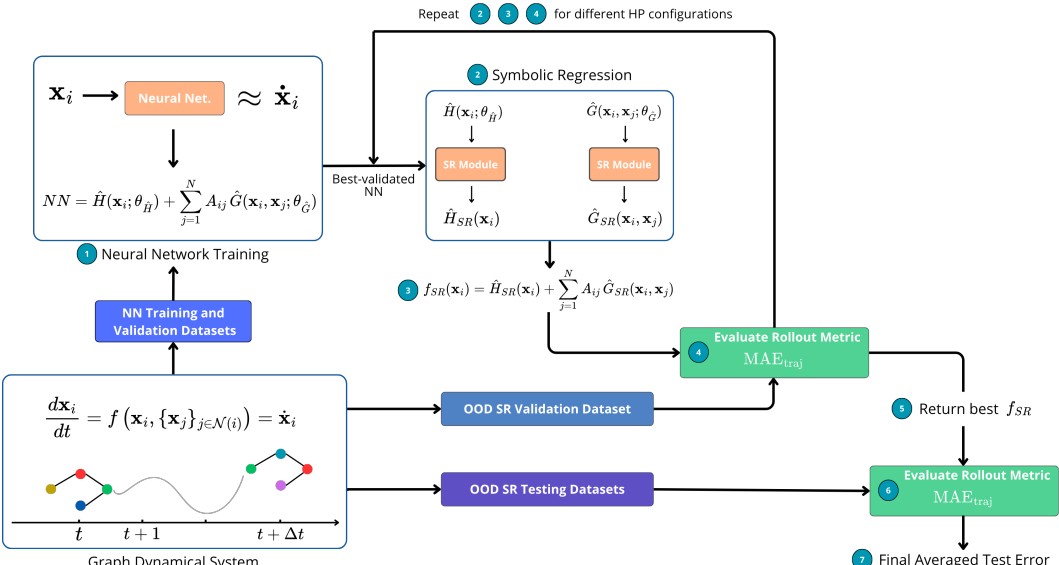

Figure 3: Overview of the experimental pipeline for model training, symbolic distillation, and evaluation. The process is sequential and designed to ensure rigorous validation of the discovered equations on out-of-distribution (OOD) data. The numbered steps are as follows: given a synthetic graph dynamical system generator, (1) a neural network architecture (e.g., GKAN-ODE, GMLP-ODE, LLC) is trained on one graph instance (trained on the first 80%, validated on the rest) to learn the underlying dynamics. (2) The trained and best-validated neural network is then used for symbolic knowledge distillation of its underlying components, $H$ and $G$, as in Eq. 3. This can be a model-agnostic, black-box approach like Genetic Programming (GP) that approximates the model's input-output behavior, or a structure-aware, white-box approach like our Spline-Wise (SW) method for KANs. (3) The candidate symbolic equations generated by the SR modules move to the evaluation phase. (4) For model selection, the symbolic formulas are evaluated on a dedicated OOD validation dataset with a different graph topology. We select the formula and its corresponding SR hyperparameters that yield the best long-term trajectory rollout performance, as measured by the $\mathrm{MAE_{traj}}$ metric. (5) The single best symbolic model from the validation step proceeds to the final testing phase. (6) For the final performance assessment, the selected model is evaluated on a separate OOD testing dataset, which contains dynamics on diverse and unseen graph topologies. (7) The resulting averaged $\mathrm{MAE_{traj}}$ score on the test set serves as the definitive metric for comparing the generalization and scientific plausibility of the discovered governing laws across all methods.

## C  ADDITIONAL RESULTS AND ABLATION STUDIES

### C.1  DETAILED PERFORMANCE ANALYSIS (MAE TIME EVOLUTION)

Fig. 4 shows the test $\mathrm{MAE_{traj}}$ over time obtained by the assessed models, including both the trained neural-based architectures and the distilled symbolic expressions. We can observe that, on EPID, BIO, and POP dynamics, GKAN-based models maintain the lowest error consistently over time, while for KUR, TPSINDy achieves the best performance.

### C.2  SYMBOLIC EXPRESSIONS EXTRACTED FROM SYNTHETIC DATASETS

We show in Tab. 10 the learned symbolic expression from the four synthetic datasets by the GMLP-ODE+GP, LLC+GP, and TPSINDy methods. Despite the two neural-based models successfully extracting the correct formulas' structures, the coefficients are slightly different from the ground truth, leading to a higher $\mathrm{MAE_{traj}}$ than GKAN-ODE+GP on every dynamics, as shown in Fig. 1.

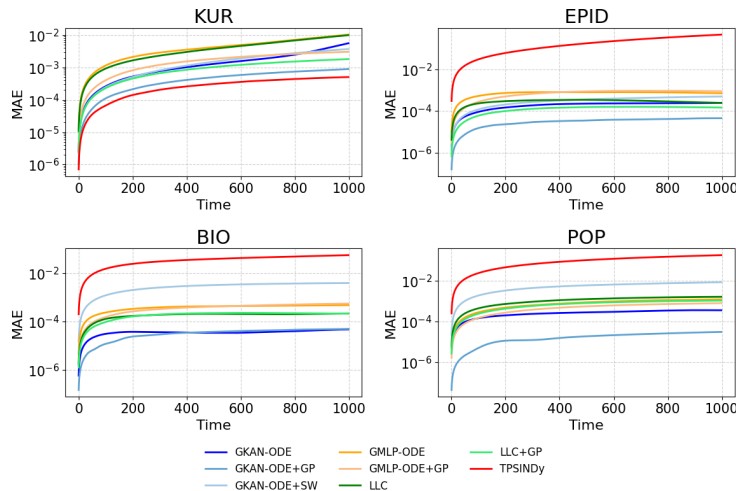

Figure 4: Evolution of test $\text{MAE}_\text{traj}$ over time for each assessed model on synthetic dynamics. Values are averaged over the three test sets.

Table 10: Learned symbolic expressions and their complexities across models and synthetic datasets.

| Model | Dataset | Learned Expression | Complexity |
|---|---|---|---|
| GMLP-ODE + GP | KUR | $2.0009 + \sum_j A_{i,j}(-0.4971 \cdot \sin(x_i - x_j))$ | 5 |
| | EPID | $-0.4990 \cdot x_i + \sum_j A_{i,j}(0.4976 \cdot x_j \cdot (1.0000 - x_i))$ | 6 |
| | BIO | $-0.4970 \cdot x_i + 0.9987 + \sum_j A_{i,j}(-0.4989 \cdot x_i x_j)$ | 6 |
| | POP | $-0.4998 \cdot x_i + \sum_j A_{i,j}(0.1973 \cdot x_j^3)$ | 5 |
| LLC + GP | KUR | $1.9995 + \sum_j(-0.4986 \cdot \sin(x_i - x_j))$ | 5 |
| | EPID | $-0.5012 \cdot x_i + \sum_j A_{i,j}(x_j \cdot (0.5005 - 0.5003 \cdot x_i))$ | 6 |
| | BIO | $-0.4971 \cdot x_i + 0.9977 + \sum_j A_{i,j}(-0.4992 \cdot x_i x_j)$ | 6 |
| | POP | $-0.4973 \cdot x_i + \sum_j A_{i,j}(0.1962 \cdot x_j^3)$ | 5 |
| TP-SINDy | KUR | $2.0000 + \sum_j A_{i,j}(0.4994 \cdot \sin(x_j - x_i))$ | 5 |
| | EPID | $-0.5679 + \sum_j A_{i,j}(0.2084 \cdot \exp(x_j - x_i))$ | 4 |
| | BIO | $0.8670 + \sum_j A_{i,j}(-0.7113 \cdot x_i x_j)$ | 4 |
| | POP | $-0.0162 + \sum_j A_{i,j}(0.0400 \cdot x_j + 0.0031 \cdot \sin(x_j))$ | 5 |

## C.3 Ablation Study: Impact of Multiplicative Nodes in GKAN-ODE

The architecture of a KAN layer with the proposed multiplicative enhancement is depicted in Fig. 5. In Fig. 6, we show the performance of GKAN-ODE models without multiplicative nodes (GKAN-ODE (no mult)) on the synthetic datasets. Despite the comparable or slightly worse $\text{MAE}_\text{Traj}$ error of GKAN-ODE(no mult), formulas extracted with GP are comparable to those obtained by GKAN-ODE. However, it is evident that the Spline-Wise fitting is unable to recognize the multiplicative term in EPID and BIO dynamics.

## C.4 Ablation Study: Comparison with Original KAN Symbolic Regression

We compare the proposed algorithm for the Spline-Wise symbolic fitting with the original one introduced by the authors of KANs. Tab. 11 shows the complexity and $\text{MAE}_\text{traj}$ of the best-validated symbolic expressions inferred by the original SW method, named GKAN-ODE+OSW. The results show that such a method is able to extract formulas that achieve very low $\text{MAE}_\text{traj}$, especially on EPID and BIO dynamics, but with very high structural complexity (thus making them less interpretable). In contrast, as shown in Fig. 1 and Tab. 2, our proposed approach consistently achieves a more favorable trade-off between accuracy and interpretability, maintaining low $\text{MAE}_\text{traj}$ while

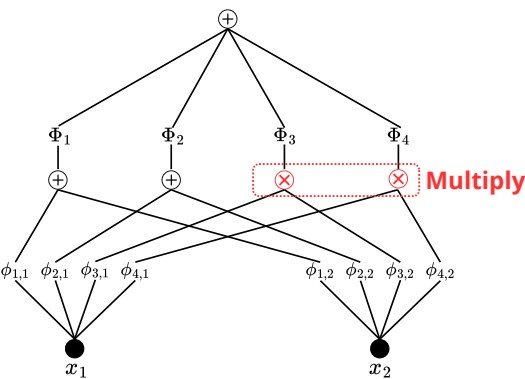

Figure 5: Representation of a KAN layer with the proposed multiplication enhancement (red) for a two-dimensional input ($d = 2$). $\phi$ are interpretable univariate splines.

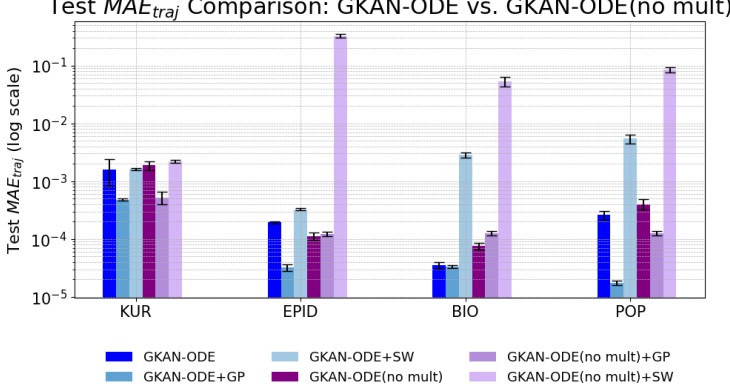

Figure 6: Performance comparison between GKAN-ODE models with and without multiplicative nodes.

yielding much more compact symbolic expressions. The hyperparameter grid employed for validating the original SW algorithm is shown in Tab. 12.

Table 11: Test $\text{MAE}_{traj}$ and structural complexity of the best-validated symbolic formulas extracted from the GKAN-ODE model. GKAN-ODE+OSW refers to the formulas obtained with the Original Spline-Wise algorithm, while GKAN-ODE+SW and GKAN-ODE+GP refer to the proposed Spline-Wise and Genetic Programming approaches. Values are averaged on three test graphs and the standard deviation is reported.

| Model | Dataset | Complexity | $\text{MAE}_{\text{traj}}$ |
|---|---|---|---|
| GKAN-ODE+OSW | KUR | 8 | $1.43 \cdot 10^{-3} \pm 2.39 \cdot 10^{-4}$ |
| | EPID | 49 | $1.40 \cdot 10^{-4} \pm 1.53 \cdot 10^{-5}$ |
| | BIO | 81 | $5.96 \cdot 10^{-5} \pm 4.82 \cdot 10^{-6}$ |
| | POP | 24 | $1.56 \cdot 10^{-2} \pm 8.28 \cdot 10^{-3}$ |
| GKAN-ODE+SW | KUR | 8 | $1.63 \cdot 10^{-3} \pm 7.67 \cdot 10^{-5}$ |
| | EPID | 10 | $3.27 \cdot 10^{-4} \pm 1.27 \cdot 10^{-5}$ |
| | BIO | 6 | $2.84 \cdot 10^{-3} \pm 3.17 \cdot 10^{-4}$ |
| | POP | 16 | $5.41 \cdot 10^{-3} \pm 1.00 \cdot 10^{-3}$ |
| GKAN-ODE+GP | KUR | 5 | $4.81 \cdot 10^{-4} \pm 2.46 \cdot 10^{-5}$ |
| | EPID | 6 | $3.22 \cdot 10^{-5} \pm 4.69 \cdot 10^{-6}$ |
| | BIO | 6 | $3.35 \cdot 10^{-5} \pm 1.91 \cdot 10^{-6}$ |
| | POP | 5 | $1.75 \cdot 10^{-5} \pm 1.44 \cdot 10^{-6}$ |

Table 12: Hyperparameter grid of the original Spline-Wise fitting algorithm.

| Hyperparameter | Values |
|---|---|
| Spline pruning threshold $\rho$ | $\{0.01, 0.05, 0.1\}$ |
| Grid range | $\{(-10, 10), (-5, 5)\}$ |
| Weight simple | $\{10^{-5}, 0.3, 0.7, 0.9\}$ |
| Symbolic library $\mathcal{F}$ | `[identity, square, cube, exp, abs,` `sin, cos, tan, tanh, ln, zero ]` |

## C.5 ROBUSTNESS ANALYSIS: OBSERVATIONAL NOISE

In the data-generating process, independent Gaussian noise is added to the node states at each time step under three different signal-to-noise ratio (SNR) levels expressed in decibels (dB): 70 dB, 50 dB, and 20 dB. The performance of the assessed models in this setting is depicted in Fig. 7. The methods are robust to noise up to 50 dB of SNR, particularly neural models, even though the quality of the expressions degrades with increasing levels of noise. This is further exacerbated by the fact that we are estimating numerical derivatives, which are highly sensitive to noise and can amplify small fluctuations in the data.

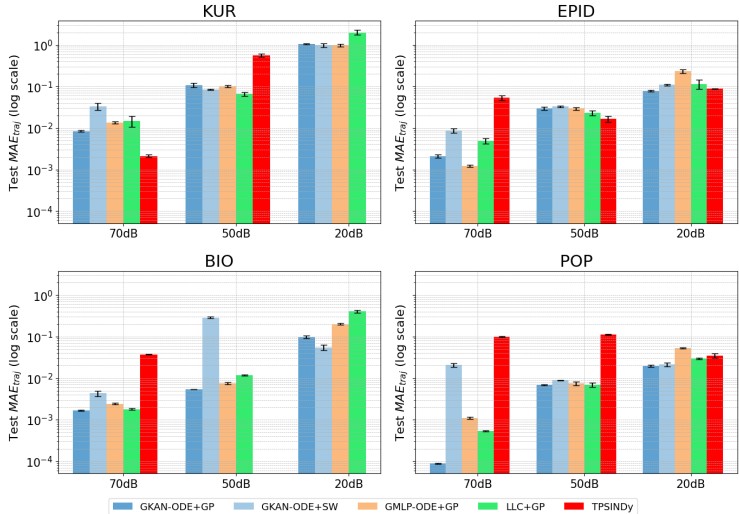

Figure 7: Performance of the extracted symbolic expression across various levels of SNR for each synthetic dataset. Missing values of TPSINDy are due to numerical divergences.

Furthermore, we adopted a more systematic anti-noise mechanism following the methodology proposed by Rudy et al. (2017). Specifically, rather than computing derivatives directly on the noisy observations, we perform a local polynomial interpolation of order $P = 3$ on the node states $\mathbf{x}(t)$. The time derivatives $\dot{\mathbf{x}}(t)$ are then computed from these smoothed polynomial proxies. This approach acts as a low-pass filter, preserving the underlying dynamics while suppressing the noise that typically destabilizes equation discovery algorithms. We evaluated this mechanism on the BIO dataset under the same signal-to-noise ratio conditions used in the main analysis. The results, reported in Table 13, demonstrate that this preprocessing step effectively stabilizes the performance of neural-based architectures. All neural models combined with symbolic regression (GP or SW) maintain trajectory errors in the order of $10^{-3}$ even at high noise levels (20 dB). In contrast, the baseline TPSINDy fails to recover accurate dynamics, exhibiting errors an order of magnitude higher ($10^{-2}$), further highlighting the superior robustness of the proposed neural-symbolic pipeline in processing noisy data.

Table 13: Test MAE$_{\text{traj}}$ (Mean $\pm$ Std) on the BIO dataset with noisy inputs, utilizing 3rd-order polynomial interpolation for robust derivative estimation.

| Model | 70 dB | 50 dB | 20 dB |
|---|---|---|---|
| GKAN-ODE+GP | $3.62 \times 10^{-3} \pm 2.25 \times 10^{-4}$ | $1.24 \times 10^{-3} \pm 1.55 \times 10^{-5}$ | $3.45 \times 10^{-3} \pm 3.11 \times 10^{-4}$ |
| GKAN-ODE+SW | $1.56 \times 10^{-3} \pm 2.22 \times 10^{-4}$ | $2.59 \times 10^{-2} \pm 2.89 \times 10^{-3}$ | $2.18 \times 10^{-3} \pm 3.21 \times 10^{-4}$ |
| GMLP-ODE+GP | $1.45 \times 10^{-3} \pm 1.59 \times 10^{-4}$ | $1.56 \times 10^{-3} \pm 1.10 \times 10^{-4}$ | $1.94 \times 10^{-3} \pm 1.40 \times 10^{-4}$ |
| LLC+GP | $1.34 \times 10^{-3} \pm 2.28 \times 10^{-4}$ | $1.66 \times 10^{-3} \pm 2.60 \times 10^{-4}$ | $2.74 \times 10^{-3} \pm 1.91 \times 10^{-4}$ |
| TPSINDy | $8.13 \times 10^{-2} \pm 1.97 \times 10^{-2}$ | $8.20 \times 10^{-2} \pm 5.02 \times 10^{-3}$ | $8.90 \times 10^{-2} \pm 4.70 \times 10^{-3}$ |

### C.6 ROBUSTNESS ANALYSIS: DERIVATIVE ESTIMATION METHOD

To ensure that the superior performance of GKAN-ODE models is not an artifact of the specific numerical differentiation technique employed (i.e., the five-point stencil method), we conducted an ablation study using the Central Finite Difference method. We focused this analysis on the BIO dataset to evaluate model sensitivity to the quality of the target derivatives $\dot{\mathbf{X}}(t)$. The results, presented in Tab. 14, demonstrate the robustness of our proposed approach. While the absolute magnitudes of the MAE$_{\text{traj}}$ shift slightly due to the lower approximation order of the central difference method compared to the five-point stencil, the relative ranking of the models remains consistent with the main experimental results. Specifically, the GKAN-ODE framework (both the neural model and the

distilled symbolic forms) continues to achieve the lowest trajectory error, consistently outperforming GMLP, LLC, and TPSINDy baselines. The SW symbolic model derived from GKAN-ODE achieves a lower error ($7.42 \times 10^{-4}$) than the black-box symbolic expression extracted from the MLP counterpart ($1.00 \times 10^{-3}$). Finally, TPSINDy continues to exhibit significantly higher error ($2.54 \times 10^{-2}$), confirming its struggle with long-term stability in this setting. These findings suggest that the performance gains reported in this work are driven by the GKAN-ODE architecture's ability to correctly capture the underlying graph dynamics, rather than sensitivity to the data pre-processing pipeline.

Table 14: Performance comparison on the BIO dataset using the Central Finite Difference method for derivative estimation. The GKAN-ODE models maintain their superior performance ranking, with the interpretable GKAN-ODE+SW outperforming the GMLP-ODE+GP baseline.

| Model | Test MAE$_{\text{traj}}$ |
|---|---|
| GKAN-ODE | $8.49 \cdot 10^{-5} \pm 8.65 \cdot 10^{-6}$ |
| GKAN-ODE+GP | $3.74 \cdot 10^{-5} \pm 1.35 \cdot 10^{-6}$ |
| GKAN-ODE+SW | $7.42 \cdot 10^{-4} \pm 9.79 \cdot 10^{-5}$ |
| GMLP-ODE | $3.61 \cdot 10^{-4} \pm 1.75 \cdot 10^{-5}$ |
| GMLP-ODE+GP | $1.00 \cdot 10^{-3} \pm 8.34 \cdot 10^{-5}$ |
| LLC | $2.33 \cdot 10^{-4} \pm 9.61 \cdot 10^{-6}$ |
| LLC+GP | $1.41 \cdot 10^{-4} \pm 2.98 \cdot 10^{-5}$ |
| TPSINDy | $2.54 \cdot 10^{-2} \pm 3.00 \cdot 10^{-3}$ |

# D SUPPLEMENTARY INFORMATION FOR REAL-WORLD EPIDEMIC DYNAMICS

## D.1 DISCOVERED EQUATIONS FOR EPIDEMIOLOGICAL SPREADING

Table 15: Symbolic expressions extracted from COVID-19 data as global dynamics, before country-specific fine-tuning. The LLC equation is re-derived from scratch, as the original work does not report all necessary coefficients needed for reproduction.

| Model | Discovered Symbolic Expression | Complexity |
|---|---|---|
| TPSINDy | $a \cdot x_i + b \cdot \sum_j A_{ij} \, 1/(1 + e^{-(x_j - x_i)})$ | 7 |
| LLC+GP | $a \cdot \tanh(x_i + b) + c \cdot \sum_j A_{ij}((x_i - x_j) \cdot e^{-x_j})$ | 7 |
| GMLP-ODE+GP | $a \cdot \ln(x_i + b) + \sum_j A_{ij} \ln(\tan(x_i + c)^2 + d)$ | 9 |
| GKAN-ODE+GP | $ax_i + b + \sum_j A_{ij}(c \cdot e^{x_j})$ | 5 |
| GKAN-ODE+SW | $a \cdot \tanh\big(b \cdot \tanh(cx_i + d) + e\big) - f \cdot \tanh\big(gx_i^3 + hx_i^2 - ix_i - j\big) + k$ $+ \sum_j A_{ij}\big(l \cdot \tanh(mx_i - n) - o \cdot \tanh(px_j - q) - r\big)$ | 30 |

## D.2 PROTOCOL FOR COUNTRY-SPECIFIC COEFFICIENT FINE-TUNING

To account for the heterogeneity of real-world epidemic dynamics, we fine-tune the coefficients of the generic symbolic structures discovered by neural-based models (detailed in Tab. 15) for each node. Specifically, we replace scalar constant terms in the symbolic equations with trainable parameters and optimize them via gradient descent. The optimization is performed by retraining the expressions on each node's data using the first 80% of observations, with the subsequent 10% for validation, and leaving the final 10% for testing. Note that the LLC equation (and subsequent fine-tuning) is re-derived from scratch, as the original work does not report all the necessary coefficients needed for reproduction. Instead, the TPSINDy formula is the one provided in the original paper. However, for a fair comparison with neural-based equations, we re-executed the fine-tuning algorithm used in the TPSINDy paper only on the first 90% of observations. This leads to a set of

coefficients very similar to the original one, but which does not depend on the entire dataset, which is crucial when evaluating the generalization capabilities of a ML model.

## D.3 ADDITIONAL TRAJECTORY PLOTS

In Figs. 8 and 9, we show the performance of the learned symbolic expressions on COVID-19 data from Canada, Brazil, Turkey, and Serbia. Fig. 8 presents the predicted trajectories obtained through autoregressive integration, while Fig. 9 illustrates the results from short-term integration. As suggested by the performance comparison depicted in Fig. 2, neural-based models are able to capture the epidemiological spreading in both scenarios, while TPSINDy struggles in long term predictions.

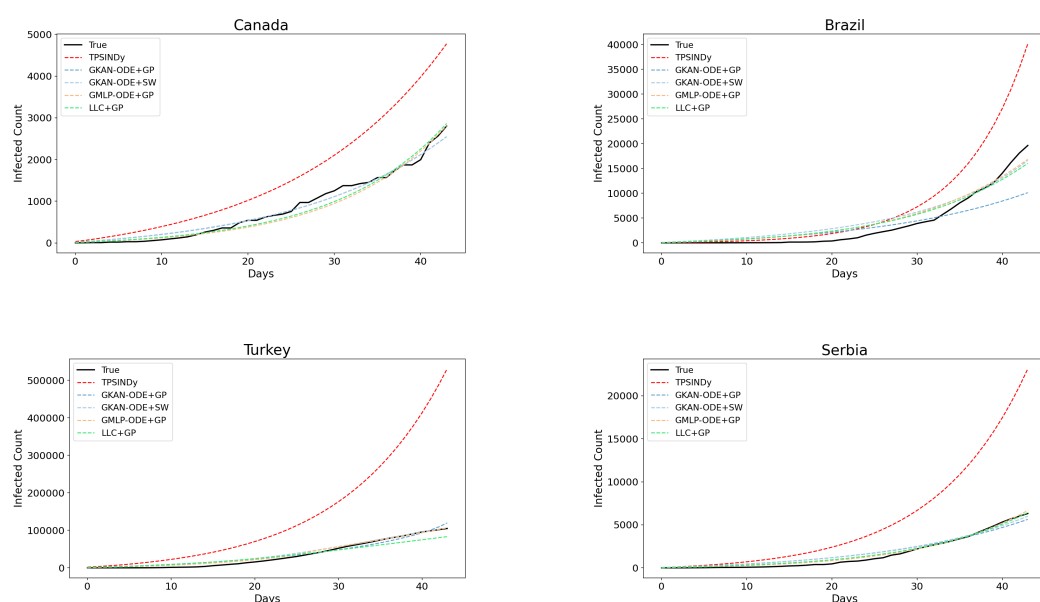

Figure 8: Predicted trajectories obtained by the long term (autoregressive) integration of the learned equations on COVID-19 data of Canada, Brazil, Turkey and Serbia.

## E DECLARATION ON GENERATIVE AI

The author(s) have employed Generative AI tools for proofreading and improving the readability of figures and tables. No LLM was involved in the research design, experiments, analysis, or in the generation of any scientific content.

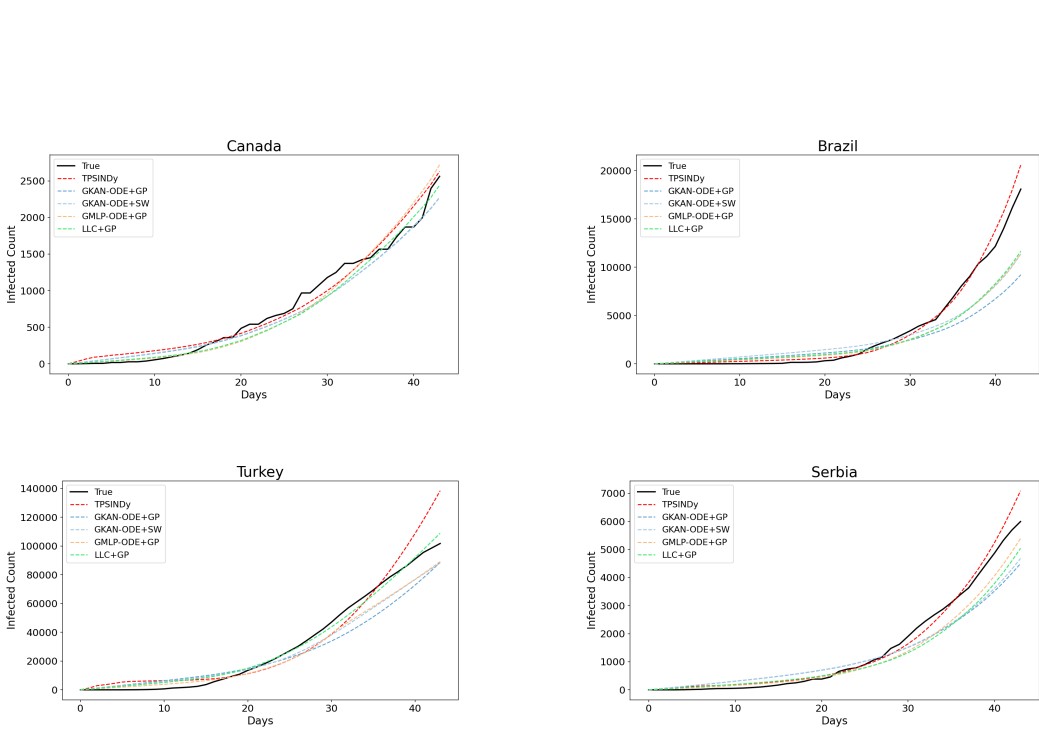

Figure 9: Predicted trajectories obtained by the short term integration of the learned equations on COVID-19 data of Canada, Brazil, Turkey and Serbia.

