# OpenReview forum: "Discovering Generalizable Governing Equations for Graph Dynamical Systems with Interpretable Neural Networks"
_ICLR.cc/2026/Conference — Submitted to ICLR 2026_

### Official Review · Reviewer_jmyg · 2025-10-20

**Soundness:** 3
**Presentation:** 3
**Contribution:** 3
**Rating:** 6
**Confidence:** 1

**Summary:**

This paper addresses the challenge of discovering symbolic governing equations for graph dynamical systems directly from observational data. The authors propose a novel architecture, the Graph Kolmogorov-Arnold Network (GKAN-ODE), which adapts KANs for this task by modeling self-dynamics and interaction dynamics separately. The model is enhanced with internal multiplicative nodes to better capture physical interactions. To extract human-readable formulas, the paper introduces a principled, structure-aware "Spline-Wise" (SW) symbolic regression algorithm. The authors establish a rigorous benchmark, evaluating GKAN-ODE and other baseline methods on synthetic and real-world epidemic datasets. The core findings demonstrate that neural-based models, particularly GKAN-ODE, significantly outperform sparse regression methods in long-term stability and generalization to unseen graph topologies, successfully recovering ground-truth equations while being more parameter-efficient.

**Strengths:**

1. Comprehensive and Rigorous Benchmarking. The paper establishes a high-quality benchmark that addresses a clear gap in the literature.
2. Novel and Well-Motivated Method. The proposed GKAN-ODE framework is technically sound and introduces several valuable innovations.
3. Principled Symbolic Distillation and Interpretability Analysis. The paper offers a thoughtful approach to extracting and analyzing symbolic models.

**Weaknesses:**

I am not familiar with this field, and my expertise is insufficient to critique the article or offer suggestions.

**Questions:**

NA

---

> ### Author Response · Authors · 2025-11-21
> **Reply to Reviewer jymg**
>
> We **gratefully thank Reviewer jmyg** for their time and for their transparency regarding their field of expertise. We are delighted that, despite this being outside your primary domain, you clearly **recognized and appreciated the core strengths of our work**. Therefore, we find your feedback **particularly encouraging**: the fact that the contribution was clear to a non-specialist suggests that our manuscript successfully communicates these complex topics to the broader machine learning community.

---

### Official Review · Reviewer_s6hu · 2025-10-30

**Soundness:** 3
**Presentation:** 3
**Contribution:** 3
**Rating:** 6
**Confidence:** 4

**Summary:**

The paper focuses on the task of discovering governing equations in dynamical systems on networks.
In particular:
- It introduces GKAN-ODE, a method based on KAN, adapted for graph data (with static topology), and where the authors introduced multiplicative nodes.
- It introduces a symbolic regression algorithm, SW, that extracts a symbolic equation from a trained KAN.
- It compares GKAN-ODE with other methods, namely TPSINDy, LLC, and a method based on MLPs, GMLP-ODE.
- In doing so, it offers a benchmark to study and compare such methods.

They argue that GKAN-ODE, by exploiting the interpretability naturally inherent in the KAN, offers an accurate and transparent method for studying complex systems.

**Strengths:**

**Originality**

As far as I know, the new elements introduced in this paper are indeed original:
- the GKAN network, with multiplicative nodes,
- and the spline-wise symbolic regression algorithm to retrieve the symbolic equation from the trained model.

**Quality**

The methodology and evaluation are sound, with no overemphasised claims, and good support of ablations and experiments over diverse methods and datasets.

**Clarity**

This is one of the clearest papers I have read in a while: the discussion flows in a very logical way, and I haven't spotted any typos.

**Significance**

The studied problem is interesting and relevant. This work offers a solid and sound overview of some state-of-the-art methods, even without considering the authors' own contribution on the topic, this element would already make it significant.

**Weaknesses:**

Here I list some concerns I have, in no particular order:

1. As stated already in the title, this work focuses on discovering governing equations for graph dynamical systems. I believe a better effort could have been made to clarify how much more difficult this task is compared to the case with no graph.
2. This work tries to find a difficult balance between proposing a new method and offering a benchmark of existing ones. In some parts, one has the impression that the authors used this approach to counter the limited results achieved on their proposed method.
3. In relation to point 2 above, it's a bit unclear what the advantage of the proposed spline-wise symbolic regression algorithm is. By looking at the results in Figure 1 (left), it's difficult to advocate for its use instead of GP. Table 2 also doesn't show encouraging results, both when compared with Table 1 and with Table 12.
4. The authors argue that the cost above is balanced by a "more direct", "more faithful", "granular", "fully transparent" view of what the model learned. I understand that interpretability is tricky to measure, but I think these points should be better supported. In the field of discovering governing equations, it seems that the only way is to check if the discovered equations align with the ground truth, and the results seem to suggest that SW has the worst alignment. It doesn't help that the comparison to the original SR method by KAN's authors has only been included in the appendix.
5. The addition of multiplicative nodes is justified solely via experiments.

**Questions:**

Referring directly to the weaknesses listed above:
1. Paragraph 4.2 shows that you validated the different methods also by changing the topology of the datasets. How did you do it? Could you maybe add to your benchmark methods that don't make use of the graph at all?
3. In paragraph 5.2, you argue that the coefficients obtained with SW are "*slightly* different", or that you get "additional *small*-coefficient terms", but, given that SW achieves worse MAE, can we really say that these errors are small?
4. Concerning the interpretability advantage of KAN+SW:
  - I recommend expanding on why this approach improves interpretability and giving more substance to the above claims.
  - Considering how the interpretability point was made, it seems that there is an overlap: the output of the method serves both to measure its performance and also to justify how interpretable the model itself is. Usually, in XAI, one measures the interpretability of a model by relating the output to the input. Is a similar approach viable here? For example, by linking the presence of some symbolic terms in the discovered governing equation to some features of the input?
  - I recommend adding the results of Table 10, where they lie without a direct comparison, to Figure 1.
5. The addition of multiplicative nodes is justified solely via experiments. Could you offer a theoretical justification too?



Two extra questions:

6. You use the five-point stencil method to build the time derivative of the trajectories. Have you performed an ablation on this? Since the $\text{MAE}_\text{traj}$ metric depends only on the trajectory itself, not on its derivative, I was wondering how robust these methods are when changing the way to estimate the derivative.
7. Have you considered using a KAGNN (*Bresson et al.*) in (2) instead of a simple KAN?

---

> ### Author Response · Authors · 2025-11-21
> **Reply to Reviewer s6hu (part 1)**
>
> We **gratefully thank** the anonymous reviewer for their appreciation of our contribution, its **praise on quality and clarity**, and the time invested and thoughtful comments to help us improve our manuscript! We address the weaknesses and questions in the following:
>
>
> **W1&Q1 (Task complexity)**: This point is relevant to assess the challenges of this task, which is further complicated by the graph topology. *Barrat et al.* (2008) analyse dynamical systems on graphs from a mathematical and statistical mechanics perspective, and show how one can model analytically their behavior only on some specific cases with strong assumptions (like mean-field), hence inferring their equation from data becomes highly non trivial, if not feasible at all. The point is that the dynamics are fundamentally driven by interactions; without knowing or considering the whole graph structure, the system is effectively partially observed, making accurate long-term prediction impossible.
> We embrace the reviewer’s suggestion, and to quantitatively demonstrate the necessity of incorporating graph topology and to clarify the difficulty gap between graph-based and standard dynamical systems, we implemented a new baseline, **MLP-ODE**.
> The MLP-ODE attempts to learn the time derivative using **only** the node's own state as input, effectively ignoring all neighbor interactions and the graph topology. We trained this model using the exact same hyperparameters, optimizer, and data splits as the graph-aware models. We have added these results to the benchmark (see updated Figure 1). The results provide strong evidence for the complexity of the task: The MLP-ODE fails catastrophically compared to GKAN-ODE. On the synthetic datasets, the error increases by orders of magnitude (e.g., on KUR: $0.408$ for MLP-ODE vs $0.0016$ for GKAN-ODE). Interestingly, the "no-graph" MLP-ODE outperforms TPSINDy on the EPID ($0.075$ vs $0.194$) and POP ($0.040$ vs $0.099$) datasets. This highlights a critical nuance: TPSINDy, although it is developed for graphs, it is restricted by a fixed library and unable to capture the correct interaction terms. In contrast, the MLP-ODE, while lacking topological information, likely learns a "safe" approximation that seems more stable than a diverging equation.
> Concerning the different graph topologies under study, more details are reported in Appendix B, and our framework aims to be robust for different families of structures, as they play an equally important role as the system’s dynamics and initial conditions.
> This all shows that the task is not merely about fitting a curve, but about disentangling self-dynamics from complex, topology-dependent neighbor interactions: a task where GKAN-ODE excels and where ignoring the graph renders the problem unsolvable.
>
> | Model | KUR ($MAE_{traj}$) | EPID ($MAE_{traj}$) | BIO ($MAE_{traj}$) | POP ($MAE_{traj}$) |
> | :--- | :---: | :---: | :---: | :---: |
> | **GKAN-ODE** | **1.61e-3** | **1.94e-4** | **3.52e-5** | **2.62e-4** |
> | GKAN-ODE+GP | 4.81e-4 | 3.23e-5 | 3.35e-5 | 1.75e-5 |
> | GKAN-ODE+SW | 1.63e-3 | 3.27e-4 | 2.84e-3 | 5.41e-3 |
> | GMLP-ODE | 4.57e-3 | 7.21e-4 | 3.84e-4 | 7.95e-4 |
> | GMLP-ODE+GP | 1.73e-3 | 7.29e-4 | 3.75e-4 | 4.96e-4 |
> | LLC | 4.20e-3 | 2.89e-4 | 1.82e-4 | 1.12e-3 |
> | LLC+GP | 9.93e-4 | 1.29e-4 | 1.88e-4 | 7.10e-4 |
> | **MLP-ODE (No Graph)** | **0.408** | **0.075** | **0.166** | **0.040** |
> | MLP-ODE+GP | 0.396 | 0.103 | 0.145 | 0.039 |
> | TPSINDy | 2.91e-4 | 0.194 | 0.036 | 0.099 |
>
> **W2 (Balance between Method vs. Benchmark)**: We acknowledge the reviewer's observation regarding the balance of the paper. We found that evaluating our proposed GKAN method was challenging due to the lack of standardized benchmarks, limited reproducibility of prior work, and superficial analysis in the existing literature. Therefore, we aimed to contribute to the community by providing a **rigorous, transparent, and fair benchmark** for symbolic equation discovery on graph dynamical systems. While this might seem to dilute the focus on the novel architecture, we argue that a strong benchmark is a prerequisite for validating *any* new method in this field. The task of Symbolic Regression is inherently complex with no "one-size-fits-all" solution, a sentiment echoed by recent initiatives like SRBench (La Cava et al., 2021) and critiques on reproducibility in scientific discovery (McGreivy et al., Nature Machine Intelligence, 2024). We believe our results are not limited, but rather honest representations of SOTA models under rigorous experimental conditions.

---

> > ### Author Response · Authors · 2025-11-21
> > **Reply to Reviewer s6hu (part 2)**
> >
> > **W3 & W4 (Advantage of SW vs. GP)**: Thank you for this crucial question. The distinction between the GP-based approach (using PySR) and our proposed Spline-Wise (SW) approach is fundamental:
> > - GP (Black-Box): This is a *surrogate* approach. It seeks the most parsimonious equation that approximates the input-output behavior of the trained neural network. It is excellent for hypothesis generation but treats the model as an opaque oracle.
> > - SW (White-Box): This approach is *intrinsic*. It translates the actual learned weights (more properly, the splines) of the KAN into symbols. It is not an approximation of behavior, but a **faithful representation of the model's internal logic**.
> > While GP achieves lower MAE by smoothing out irregularities, SW provides trustworthiness by revealing exactly what the network learned, including its imperfections. Comparing Tables 1 and 2 shows that SW recovers the correct structural forms. The differences lie in coefficient precision (e.g., 0.4988 vs 0.5), which leads to the higher error variance in Figure 1. However, this transparency is a unique feature of GKANs that MLPs (and thus GP on MLPs) cannot offer. Furthermore, as detailed in the improved Appendix C.4 (Tab 11), our proposed SW algorithm significantly outperforms the original KAN SW implementation in balancing complexity and accuracy, marking a clear methodological improvement.
> >
> > **W5 & Q5 (Multiplicative nodes justification)**: The justification is both empirical and theoretical. Theoretically, while the Kolmogorov-Arnold theorem states that sums of univariate functions are universal approximators, learning multiplicative interactions (e.g., $x \cdot y$) via addition requires learning logarithmic compositions ($e^{\ln x + \ln y}$) or binomial expansion. This is notoriously difficult for neural networks to learn with limited data and shallow architectures, as the functions are complex and undefined at zero. While *KAN 2.0* (Liu et al., 2024) proposed explicit multiplication nodes, they require fixing the quantity of such nodes as a hyperparameter, complicating model selection. Our approach introduces these nodes without additional hyperparameters, allowing the sparsity regularization to prune them if unnecessary. This provides the necessary inductive bias for physical systems (which often involve products) while maintaining the flexibility of the architecture, as confirmed by the superior performance in our ablation study (Appendix C.3).
> >
> >
> > **Q2 (Small coefficient errors)**: You are correct that "small" is relative. In Table 2, errors such as finding a coefficient of $0.4988$ instead of $0.5$ (an error of $<0.3\%$) or finding a spurious term like $0.0022 \cdot x_i$ seem negligible. However, in dynamical systems (particularly chaotic ones) these minor deviations accumulate exponentially over time. This explains why the $MAE_{traj}$ is higher for SW than for GP, despite the symbolic structure being nearly identical. We argue that these errors are "small" in the sense that the *scientific discovery* process has succeeded: the correct physical law structure has been identified, and the parameters are close enough to be refined by standard parameter estimation techniques once the structure is known. We have clarified this distinction in the revised text.
> >
> > **Q3 (Interpretability/XAI)**: We have expanded the text to clarify that interpretability here means the ability to inspect the model's components *before* aggregation. Unlike post-hoc XAI (like SHAP/LIME) which approximates local behavior, SW converts the global model into a human-readable format. In our context, the "feature importance" is explicitly given by the symbolic terms. If a term $x_j^3$ appears in the equation, it is a global assertion that the cubic of the neighbor's state drives the dynamics. This is a stronger form of XAI than attribution scores. Finally, we have updated the Appendix and discussion with results of the "Original SW" method from Table 10 (now 11).

---

> > > ### Author Response · Authors · 2025-11-21
> > > **Reply to Reviewer s6hu (part 3)**
> > >
> > > **Q6 (Derivative estimations)**: To assess whether our results are artifacts of the specific derivative estimation method (i.e., five-point stencil), we conducted an ablation study using the Central Finite Difference method on the BIO dataset (with other datasets currently running, and will be provided as soon as possible). The results, reported in the table below (and added to Appendix C.6), confirm the robustness of our approach. Even **with this different estimation technique, the relative performance ranking of the models remains consistent** with the main paper:
> > > GKAN-ODE architectures continue to achieve the lowest $\text{MAE}_{\text{traj}}$, significantly outperforming TPSINDy ($2.54 \times 10^{-2}$) and GMLP/LLC baselines. Notably, the **GKAN-ODE+SW** ($7.42 \times 10^{-4}$) outperforms the black-box **GMLP-ODE+GP** ($1.00 \times 10^{-3}$) in this setting. While the absolute error values shift slightly due to the lower precision of central differences compared to the five-point stencil, the GKAN-ODE+GP remains the top-performing symbolic model ($3.74 \times 10^{-5}$). This all seems to confirm that the superiority of GKAN-ODE is driven by the architecture's ability to capture graph dynamics, rather than sensitivity to this pre-processing step.
> > >
> > > | Model | Test $\text{MAE}_{\text{traj}}$ | Std |
> > > | :--- | :---: | :---: |
> > > | **GKAN-ODE** | $8.49 \times 10^{-5}$ | $8.65 \times 10^{-6}$ |
> > > | **GKAN-ODE+GP** | $3.74 \times 10^{-5}$ | $1.35 \times 10^{-6}$ |
> > > | **GKAN-ODE+SW** | $7.42 \times 10^{-4}$ | $9.79 \times 10^{-5}$ |
> > > | GMLP-ODE | $3.61 \times 10^{-4}$ | $1.75 \times 10^{-5}$ |
> > > | GMLP-ODE+GP | $1.00 \times 10^{-3}$ | $8.34 \times 10^{-5}$ |
> > > | LLC | $2.33 \times 10^{-4}$ | $9.61 \times 10^{-6}$ |
> > > | LLC+GP | $1.41 \times 10^{-4}$ | $2.98 \times 10^{-5}$ |
> > > | TPSINDy | $2.54 \times 10^{-2}$ | $3.00 \times 10^{-3}$ |
> > >
> > > **Q7 (KAGNN)**: Thank you for bringing up the work of Bresson et al. (TMLR 2025). As briefly mentioned in Section 2.3, their work implements KAN-based GNN layers (KAGCN, KAGAT, KAGIN) primarily for graph classification and regression tasks, which differs from our objective of discovering continuous-time governing equations. Specifically, our architecture models the time derivative $\dot{\mathbf{x}}_i =  H(\mathbf{x}_i) + \sum_{j=1}^N A_{ij} \, G(\mathbf{x}_i, \mathbf{x}_j)$, where $H$ and $G$ are distinct KANs representing self and interaction dynamics. In contrast, Bresson et al. typically aggregate neighbors *before* or *during* the KAN application (e.g., their KAGIN uses $\text{KAN}(\mathbf{x}_i + \sum \mathbf{x}_j)$). While we cannot directly use their architecture for our decoupled ODE formulation, the shared intuition of replacing MLPs with KANs enhances performance and interpretability, is validated by both works. Our work further extends this intuition specifically into the realm of scientific discovery and symbolic extraction.
> > >
> > > We hope that **our response has satisfactorily addressed your questions**. If you find that our clarifications resolve your concerns, we would be grateful for a reconsideration of the score. We remain happy to discuss any remaining points.

---

> > > > ### Comment · Reviewer_s6hu · 2025-11-27
> > > > **Satisfied by authors' replies**
> > > >
> > > > I sincerely thank the authors for their careful and detailed answers, and overall for the interesting read :)
> > > >
> > > > All my concerns were addressed, in particular:
> > > > - The importance of the topology is now clear.
> > > > - The choice of the structure of the paper, midway between a benchmark and the proposal of a new method, seems justified.
> > > > - The authors added a deeper discussion clarifying the role of interpretability in the context of discovering governing equations and the comparison between SW and GP. I see that other reviewers raised a similar concern, but I believe that the discussion about this point can be brought up in a wider arena, with the whole community, instead of limiting it to this forum.
> > > > - Ablations on the multiplicative nodes and on the derivative estimation method have been added.
> > > >
> > > > Due to the above, I will increase the score of my review.
> > > >
> > > > Good luck!

---

> > > > > ### Author Response · Authors · 2025-11-28
> > > > >
> > > > > We gratefully thank reviewer s6hu for their thoughtful follow-up and for taking the time to reassess their review and increase the score. We truly appreciate your careful reading of our work and the constructive suggestions. We are glad to hear that the revisions addressed your concerns and we would be enthusiastic to carry the dialogue on interpretability and equation discovery forward in a wider community setting.
> > > > >
> > > > > Thank you again for your engagement and kind wishes.

---

### Official Review · Reviewer_K8qq · 2025-10-31

**Soundness:** 3
**Presentation:** 2
**Contribution:** 2
**Rating:** 2
**Confidence:** 5

**Summary:**

This work focuses on graph dynamical systems (GDS) and proposes a novel GKAN-ODE framework that integrates Kolmogorov-Arnold Networks (KAN) with neural differential equations. Combined with an interpretable Spline-Wise symbolic regression pipeline, this framework enables the automated discovery of universal governing laws underlying graph-structured dynamical systems.

**Strengths:**

-By leveraging the spline-based structure of the KAN model and the white-box interpretability of the Spline-Wise mechanism, the proposed method successfully transforms deep network representations into interpretable symbolic expressions, which helps reveal the model’s internal learning dynamics.

-To promote reproducibility and support open science, all code and experimental configurations associated with this work have been publicly released.

**Weaknesses:**

-The current version of the paper lacks an intuitive illustration of the overall architecture. A well-structured framework figure would greatly enhance the presentation by clarifying the relationships and information flow among different components of the proposed model.

-The core evaluation relies on numerical integration to obtain X^(t) and compute MAEtraj(Eqs. 5–6). Since this process is autoregressive, the accumulated error may vary with the integration step size and the choice of integrator. However, the paper does not provide an analysis of integrator robustness, leaving the stability and reliability of the results insufficiently validated.

-The authors propose "multiplicative nodes without hyperparameters" to enhance physical interaction modeling, but they lacked a visual analysis of why it is superior to LLC/GMLP.

-The noise robustness analysis is not in-depth enough. The experimental design includes noise versions of different SNRS, but the systematicness of the derivative estimation and anti-noise mechanism in the paper is relatively limited.

-From the experimental results in Figure 1, it can be seen that the GKAN-ODE-SW method is not as good as GKAN-ODE-GP, failing to demonstrate the effectiveness of the proposed method in handling graph dynamical systems.

**Questions:**

Q1：To what extent does the long-term rollout error depend on the choice of numerical integrator and step size? While the paper reports overall evaluation results, why are the detailed experimental settings—such as the integrator type, step size, or tolerance—omitted?

Q2：Given that the primary metric MAEtraj relies on numerical integration, would different integrators (e.g., RK4, DOPRI) or step-size configurations change the ranking of model performance? Has the sensitivity of this metric to the integration method been analyzed?

Q3：In the synthetic tasks, the authors use OOD sets with varying topologies and initial conditions for selecting the final symbolic expressions, whereas in the real epidemic scenario, model selection is performed only on the training set due to the lack of OOD data. Has the potential “model selection bias” between these two settings been examined?

Q4: The authors emphasize that MAEtraj is independent of the ground-truth equations and, therefore, more stringent due to error accumulation. Could the authors provide a correlation analysis between MAEtraj and MAEeul to show which metric better aligns with scientific correctness when their results diverge?

Q5：What is the impact of the proposed KAN architecture on the experimental results? Furthermore, please provide a detailed description of its specific implementation in the paper.

---

> ### Author Response · Authors · 2025-11-21
> **Reply to Reviewer k88q (part 1)**
>
> We **gratefully thank Reviewer K8qq** for their time and for highlighting the strengths of our work, specifically acknowledging that our framework **"successfully transforms deep network representations into interpretable symbolic expressions"** and praising our commitment to **open science**. We understand your concerns regarding the presentation and robustness analysis. We have taken your feedback very seriously and conducted additional experiments and clarifications to address them.
>
> **W1 (Pipeline Illustration):** We agree with the reviewer that a visual representation improves clarity. We added a comprehensive pipeline diagram in Appendix B.2, which illustrates the full evaluation framework flow.
>
> **W2, Q1, Q2 (Numerical integrator robustness):**
> We apologize if the details regarding the integration scheme were not sufficiently prominent.
> As detailed in Appendix A.2, we employ the `dopri5` integrator from the `torchdiffeq` library. This is an adaptive-step Runge-Kutta method of order 5(4). This choice is deliberate and standard in state-of-the-art Neural ODE literature (e.g., *Chen et al., NeurIPS 2018*; *Poli et al., Graph-ODE, 2019*). The use of an **adaptive step size solver eliminates the step size as a hyperparameter**, as the solver automatically adjusts the step to maintain error within the specified tolerances (`atol=rtol=1e-5`).
> Because the model learns the continuous dynamics function $f$ (where $\dot{\mathbf{x}} = f(\mathbf{x}, A)$) rather than a discrete mapping, the choice of a high-order adaptive integrator ensures that the evaluation is robust. We assume that changing to another high-order adaptive solver (e.g., `adams`) would not significantly alter the ranking of the models, as the primary source of error is the learned vector field $\hat{f}$, not the integration scheme. The stability of our results across diverse datasets and conditions confirms this robustness, as reported in Appendix C.5 and C.6. Nevertheless, we are willing to clarify this point with more experiments if requested.
>
> **W3 (Multiplicative nodes)**: We have included a visual depiction of the GKAN architecture with multiplicative nodes in Appendix C.3.
> The superiority of our approach over LLC is both structural and empirical. LLC enforces a rigid interaction structure using three separate MLPs: $G(x_i, x_j) = \text{MLP}_1(x_i, x_j) + (\text{MLP}_2(x_i) \cdot \text{MLP}_3(x_j))$. This forces a multiplicative bias even if the dynamics are purely additive. In contrast, our GKAN approach embeds multiplicative nodes *inside* the layer alongside additive ones. Combined with sparsity regularization, the network **automatically learns** whether to use multiplication or addition, without requiring architectural hyperparameters. Our ablation study (Appendix C.3) and main results (Figure 1) confirm that this flexibility leads to better performance.
>
> **W4 (Noise robustness analysis)**: We appreciate this critique. To rigorously address the systematicness of derivative estimation under noise, we have extended our analysis. We adopted the **denoising procedure** proposed in *Rudy et al., Science Advances 2017*, which fits a polynomial of degree $P=3$ to the noisy trajectory window to estimate derivatives, rather than using raw finite differences. We have also conducted additional experiments (preliminary, due to time constraints) on the BIO dynamical system across the three noisy levels tested in the paper (70dB, 50dB, 20dB). The following table illustrates these preliminary results:
>
> | Model       | 70 dB (MAE ± STD)                   | 50 dB (MAE ± STD)                   | 20 dB (MAE ± STD)                   |
> |-------------|-------------------------------------|-------------------------------------|-------------------------------------|
> | GKAN-ODE+GP | (3.62 × 10^-3 ± 2.25 × 10^-4)       | (1.24 × 10^-3 ± 1.55 × 10^-5)       | (3.45 × 10^-3 ± 3.11 × 10^-4)       |
> | GKAN-ODE+SW | (1.56 × 10^-3 ± 2.22 × 10^-4)       | (2.59 × 10^-2 ± 2.89 × 10^-3)       | (2.18 × 10^-3 ± 3.21 × 10^-4)       |
> | GMLP-ODE+GP | (1.45 × 10^-3 ± 1.59 × 10^-4)       | (1.56 × 10^-3 ± 1.10 × 10^-4)       | (1.94 × 10^-3 ± 1.40 × 10^-4)       |
> | LLC+GP      | (1.34 × 10^-3 ± 2.28 × 10^-4)       | (1.66 × 10^-3 ± 2.60 × 10^-4)       | (2.74 × 10^-3 ± 1.91 × 10^-4)       |
> | TPSINDy     | (8.13 × 10^-2 ± 1.97 × 10^-2)       | (8.20 × 10^-2 ± 5.02 × 10^-3)       | (8.90 × 10^-2 ± 4.70 × 10^-3)       |
>
> The polynomial interpolation prevents error explosion as SNR increases, particularly for GMLP-ODE and LLC which show strong robustness when using this denoising procedure. We added a paragraph in Appendix C.5 regarding this robustness analysis. Experiments on the remaining dynamical systems (KUR, EPID, POP) are currently running and will be included as soon as they finish.

---

> > ### Author Response · Authors · 2025-11-21
> > **Reply to Reviewer k8qq (part 2)**
> >
> > **W5 (GKAN-ODE+SW vs. GKAN-ODE+GP):** We respectfully clarify that our goal is not to claim that SW outperforms black-box GP on raw MAE, but to **provide an unbiased and systematic comparison of symbolic extraction methods** and offer **interpretable-by-design** solutions for equation discovery in graph dynamical systems. PySR is indeed a powerful symbolic regressor, it smooths over the network's imperfections, often resulting in lower error but detaching from the model's internal state, and we report this transparently. SW is not a replacement for GP, but an additional interpretability tool exclusive to KAN-based models that has the advantage of providing an interpretable neuron-by-neuron decomposition that model-agnostic black-box GP methods cannot offer. As shown in Table 2, SW successfully recovers the correct symbolic *structure* (e.g., sine waves, interaction terms). The higher MAE comes from minor coefficient drift (e.g., finding 0.498 vs 0.5), which accumulates in chaotic systems. Furthermore, our experiments demonstrate that our improved SW algorithm obtains more compact symbolic expressions, closely aligning with the ground truth (see **Table 11** in Appendix C.4), with a better performance-complexity trade-off than the original KAN approach.
> > It serves as a necessary tool for scientists who need to audit exactly what the neural network learned, rather than just fitting its output.
> >
> > **Q3 (model-selection bias in real-world data)**: We agree that using in-distribution validation for the Real-World data introduces a potential bias due to the lack of a true OOD ground truth. However, we address this by testing **transferability**. We show (Section 5.3) that the symbolic formulas learned on the COVID-19 dataset, after simple coefficient fine-tuning, generalize effectively to the H1N1 and SARS datasets. This successfully demonstrates that the discovered *structure* is robust and physically plausible, mitigating concerns that the model simply memorized the training set.
> >
> > **Q4 (MAE_traj vs. MAE_eul Correlation):** We thank the reviewer for this insightful suggestion. We are currently finalizing this correlation analysis to highlight exactly when and how these metrics diverge, and which aligns better with the ground truth in limiting cases. We are working to provide these detailed results as soon as possible and sincerely appreciate your patience.
> > **Q5 (KAN implementation specifics):** A detailed description of KAN model is presented in Appendix A.1. The pseudo-code for the SW algorithm can be found in Appendix A.4, while Appendix C.3 contains a Figure illustrating multiplicative nodes. Furthermore, Appendix A.5 provides the search space for hyperparameter tuning. The whole codebase is documented, reproducible, and provided in the Supplementary Material. As it is a main component of our contribution, we plan to release it as **open source package**, hoping to foster its adoption in the community and for AI for Science.
> > Given its importance to our overall contribution, we intend to release it as an **open-source package**. We hope this will encourage its widespread adoption within the community, particularly for applications in AI for Science.
> >
> > In summary, we wish to emphasize that our contribution extends beyond proposing a single architecture; it establishes a **systematic and rigorous evaluation framework** for equation discovery on graph dynamical systems. We move the field beyond simple curve fitting by prioritizing **OOD generalization** and **autoregressive stability**, supported by the robust noise and derivative estimation analyses you requested.
> > Furthermore, the **GKAN-ODE architecture** represents a crucial methodological step, effectively **bridging the gap between KANs for equation discovery** (Koenig et al., 2024) **and KANs as graph neural networks** (Bresson et al., 2024). By introducing multiplicative nodes and an improved Spline-Wise algorithm, we demonstrate that it is possible to extract **compact, interpretable symbolic expressions** that are more faithful than black-box approximations and more parameter-efficient than MLPs.
> >
> >
> > We believe this work fills a critical gap in the literature, offering the community the tools necessary for reproducible progress. We trust that **the new experiments (denoising analysis), the pipeline visualization, and these clarifications have resolved your concerns**. We respectfully ask you to consider these significant updates in your evaluation, and we remain available for any further discussion.

---

### Official Review · Reviewer_Usk1 · 2025-10-31

**Soundness:** 2
**Presentation:** 3
**Contribution:** 2
**Rating:** 2
**Confidence:** 3

**Summary:**

This is a nicely written paper that uses a multi-step learning process and new architecture to learn symbolic differential equations. They then try to extract symbolic structure from their learned network as well as a few alternative networks. They try this approach on a few symbolic ODEs on networks to define a benchmark set.

**Strengths:**

Their goal is valuable: interpretability for graph-structured ODEs. I also appreciate that they're trying fundamentally different architectures with edge nonlinearities instead of node nonlinearities. I think their tasks are reasonable targets.

**Weaknesses:**

Overall I'm sympathetic to the concept but I have multiple major concerns.

1) I think they overstate the importance of their benchmarks. It’s a limited set of tasks. These are fine tasks, I don’t object at all, but I would not elevate them to the level of a rigorous benchmark that makes its own contribution.

2) They talk about the greater interpretability of their GKAN networks, but if I understand correctly, they are really doing a non-symbolic fit with a different architecture, then using symbolic regression to distill their model. They do the same thing for other models too, and again get symbolic expressions that have similar complexity and performance. So it’s unclear what the advantage of their approach is. Their claim is that somehow putting nonlinearities on edges rather than nodes is valuable, but I don’t see evidence for this.

3) If they’re targeting symbolic nonlinearities, why not just use symbolic nonlinearities as their basis functions, instead of using splines as intermediaries?

4) They explore a particular family of composition, sums and products. This is creeping toward both node nonlinearities, and toward directly fitting symbolic structure. So it seems like they’re trying to have it both ways: “let’s use edge nonlinearities instead of node nonlinearities — except for here.” And “let’s fit neural networks and then later find symbolic structure — except for here.”

5) I don’t understand why their Spline-wise regression is unique to KANs. They’re just ways of fitting a symbolic function to another function, and that should work for general functions as well as for splines.

Overall, this is a reasonable approach, but I’m not convinced that it’s a substantial advance. I may be mistaken about any of these points, and I’m happy to be corrected.

**Questions:**

What is the main contribution here? Am I missing a substantive performance difference or improvement in interpretability from the GKAN versus the GMLP?

---

> ### Author Response · Authors · 2025-11-21
> **Reply to Reviewer Usk1 (part 1)**
>
> We **gratefully thank Reviewer Usk1** for acknowledging the value of our goal, interpretability for graph-structured ODEs, and for appreciating our exploration of fundamentally different architectures. We value the reviewer’s skepticism regarding the specific advantages of GKANs, as it allows us to clarify the unique properties of KANs compared to standard MLPs and sparse regression. We address the major concerns below.
>
>
> **W1 (Importance of the Benchmark)**: We respectfully argue that the significance of this benchmark lies not in the raw quantity of datasets, but in the **methodological rigor and the critical analysis of the task**, addressing a **severe gap in the current literature** on equation discovery in graph dynamical systems.
> Symbolic regression is already a formidable challenge, notoriously sensitive to hyperparameter tuning and evaluation metrics on limited benchmarks, as highlighted by initiatives like *SRBench (La Cava et al., 2021)* and recent critiques on reproducibility in AI for Science *(McGreivy et al., *Nature Machine Intelligence*, 2024)*.
> Discovering governing equations for *graph* dynamical systems introduces a higher layer of complexity: the laws are not merely functions of state $x(t)$, but are driven by complex, topology-dependent interactions $\sum A_{ij} G(x_i, x_j)$.
> To quantitatively prove that these tasks are non-trivial and cannot be solved by standard SR methods, we implemented a new **graph-agnostic MLP-ODE baseline** (as suggested by Reviewer s6hu) that ignores topological information. As shown in the revised Figure 1, this baseline fails catastrophically (e.g., on Kuramoto, the error increases by two orders of magnitude compared to GKAN-ODE). This confirms that this task requires specific model development and research. Additionally, existing works (e.g., the TPSINDy evaluation, one of the few SOTA models for this task) often rely on single-step prediction error (non-autoregressive). We demonstrate that this is misleading for scientific discovery: a model can have low single-step error but diverge instantly in simulation. Our benchmark introduces a stricter protocol: a **long-term stability ($\text{MAE}_{\text{traj}}$)** evaluated via full autoregressive rollout. We show that while baselines like TPSINDy appear competitive in single steps, they suffer from error accumulation in the long term. GKAN-ODE is the first to demonstrate stable recovery of the dynamics. We also investigated **OOD generalization**, where we do not just test on held-out time steps. Lastly, we test on **entirely new graph topologies**, being a reasonable-but-neglected standard for verifying if a physical law has truly been discovered, rather than the network simply memorizing a specific graph instance.
>
> To our knowledge, this is the **first systematic comparison** of neural-based symbolic learners for graph dynamical systems. By establishing a **fair, open-source framework** with additional analysis on noise injection, derivative estimation (Appendix C), and model selection, we provide the community with the necessary guidance and tools to measure progress.
>
> In conclusion, **this work is not merely a proposal of a new architecture (GKAN-ODE), but a foundational step toward rigorous validation in AI for Science**, and we hope this benchmark serves as a catalyst, inspiring the community to adopt these **more solid and critical evaluation practices**, thereby fostering more robust and reproducible progress in the data-driven discovery of complex dynamics. We show that GKAN-ODE succeeds where sparse regression fails, not by chance, but because it is carefully evaluated under conditions that genuinely reflect the difficulty of scientific discovery.

---

> > ### Author Response · Authors · 2025-11-21
> > **Reply to Reviewer Usk1 (part 2)**
> >
> > **W2 (GKAN Interpretability)**: GKANs sit in between classical neural networks and symbolic models, and its advantage is not merely in the final performance number, but in the **structural transparency** that the architecture affords, which fundamentally changes *how* symbolic regression can be performed.
> >
> > Understandably, the reviewer questions the value of placing nonlinearities on edges, and, therefore the rationale behind the overall KAN architecture. In standard MLPs, the computational graph relies on linear transformations followed by fixed, global nonlinearities ($y = \sigma(\mathbf{W}\mathbf{x})$). Consequently, individual edge weights $w_{ij}$ are merely scalar coefficients that lack independent symbolic meaning; they effectively modulate magnitude but do not define functional form, which is given by the non-linearities and layers’ depth. This necessitates a "black-box" approach, where symbolic extraction must approximate the model's **global input-output behavior post-hoc**, for instance, with Genetic Programming.
> > Conversely, in a KAN (Liu et al., 2024), the operation is $y = \sum \phi_{i}(x_i)$. The "edges" are **learnable univariate functions** $\phi$ (parametrized as splines, but other choices have been proposed). This architecture is rooted in the Kolmogorov-Arnold representation theorem, which decomposes multivariate functions into sums of univariate compositions. This design renders KANs intrinsically interpretable ("white-box"): unlike scalars, the learned splines can be individually visualized and inspected to identify functional shapes (e.g., sinusoidal, exponential) *component-wise*, even before symbolic regression is applied. This granular inspectability is a structural capability that MLPs mathematically cannot provide. For further reference, please refer to Section 3 “KANs are Interpretable” of Liu et al, 2024.
> >
> > Finally, while we acknowledge that black-box GP often yields lower error (as it smooths over local imperfections), our **improved SW algorithm** (Table 2 and 11 in the Appendix) yields symbolic expressions that are structurally faithful to the neural model and significantly more compact than those produced by the original KAN method (see Appendix C.4), **aligning more closely to the simple ground-truth equations**. This demonstrates that the advantage of GKAN is not just theoretical, but provides an alternative pathway to discovery with AI that is more transparent than standard SR.
> >
> >
> > **W3 (Symbolic nonlinearities as KAN basis):**
> > We gratefully acknowledge this remark, as it touches upon the fundamental distinction between sparse regression methods, such as SINDy, and neural-based symbolic learning. The approach of directly utilizing symbolic nonlinearities as basis functions corresponds precisely to the methodology employed by TPSINDy, which we include as a primary baseline in our study. However, **our empirical results demonstrate that this strategy suffers from two critical limitations that GKAN-ODE overcomes**. First, basis function approaches are generally restricted to linear combinations of the provided library terms. Consequently, they struggle to capture compositional functions (such as nested nonlinearities like $\sin(\exp(x))$) unless those exact composite terms are explicitly pre-included in the library. Second, attempting to circumvent this by expanding the library to include a rich set of composite interactions leads to a combinatorial explosion in the feature space, rendering the sparse regression problem ill-posed and computationally unstable due to the curse of dimensionality. In contrast, our use of splines as intermediaries allows GKAN-ODE to function as a universal approximator that is not constrained by a fixed symbolic vocabulary during the training phase. By learning the functional shapes in a continuous and differentiable manner first, the model effectively narrows down the vast search space of mathematical expressions and can recover composite expressions only starting from univariate symbols ($\sin, \exp$). The splines act as a flexible library, allowing the network to discover complex relationships implicitly via gradient descent, which can then be distilled into precise symbolic forms in the subsequent regression step. This hybrid approach combines the expressivity of deep learning with the parsimony of symbolic regression, hence avoiding the rigid limitations of fixed basis functions.
> >
> > Simply put, one can think of initializing KAN splines as symbolic formulas, but in this way, they would lose the flexibility of splines, while hoping that the KAN initialization matches the target function.

---

> > > ### Author Response · Authors · 2025-11-21
> > > **Reply to Reviewer Usk1 (part 3)**
> > >
> > > **W4 (non-linearities):** We clarify that we are not mixing paradigms arbitrarily. The core theorem of Kolmogorov-Arnold states that multivariate functions can be represented as sums of univariate non-linearities. However, physical laws often involve products (e.g., gravity $\sim m_1 m_2$, infection rates $\sim S \cdot I$). While sums can approximate products (via log-sum-exp), it is inefficient.
> > > Our introduction of **multiplicative nodes** is a specific, parameter-free inductive bias to support physical interaction terms within the KAN framework. It does not revert to "node non-linearities" in the MLP sense (global activations); rather, it enables the network to efficiently route information for multiplicative interactions while retaining the learnable splines on the edges, showing its effectiveness and improvement over the original architecture (Appendix C.3).
> > >
> > > **W5 (SW exclusive to KANs):** The SW fitting procedure can indeed be applied to any univariate function. However, what is unique to KAN is that the specific KAN architecture (in which each neuron is a sum/product of univariate functions) gives us the information about how we can compose univariate functions to match the model’s output. In MLPs, instead, each layer applies a matrix multiplication followed by a global nonlinearity, and there is no structural decomposition into univariate functions. Therefore, SW cannot be used to reconstruct the network’s symbolic behavior.
> > >
> > > **Q1 (main contribution)**: Our main contribution is not a single architecture, but systematic and rigorous study of neural-based symbolic learners for graph-structured ODEs, including:
> > > A **rigorous and principled evaluation framework for equation discovery on graph dynamical system**, fully fair and reproducible, that considers that prioritizes OOD generalization and autoregressive stability over simple curve fitting, with a analysis on noise and derivative estimation robustness.
> > > **GKAN-ODE Architecture:** A novel adaptation of KANs, with multiplicative nodes and an improved spline-wise symbolic regression algorithm, for graph dynamics that outperforms MLPs and SINDy in accuracy and parameter efficiency (Fig. 1). **This bridges the gap between KANs in the field of equation discovery (Koening 2024) and KAN as graph neural networks (Bresson 2024)**.  We show that this leads to **more compact and interpretable symbolic expressions**, outperforming the original KANs, and being distinct from the black-box approximations of Genetic Programming.
> > >
> > > Our contribution fills this gap in the literature, which currently lacks a comprehensive and rigorous evaluation for symbolic graph ODE learning.
> > >
> > > We trust that these clarifications help resolve your questions regarding our work. We would appreciate it if you could consider reflecting these updates in your evaluation, and we look forward to any further discussion.

---

### Author Response · Authors · 2025-11-21
**General Statement**

We **gratefully thank** the anonymous **reviewers Usk1, K8qq, s6hu**, and **jmyg** for the time invested in reviewing our manuscript and for their insightful, constructive feedback. The reviews have been instrumental in identifying areas where we could strengthen our contribution and clarify the unique positioning of our GKAN-ODE framework.

A mutual concern raised across reviews involved the **robustness of our evaluation metrics** (specifically regarding noise and derivative estimation) and the **complexity of the task** itself compared to non-graph settings. To address these points and empirically verify our claims, we have conducted extensive additional analyses and revisions:

*   To quantitatively demonstrate the necessity of incorporating graph topology and to clarify the difficulty gap between graph-based and standard dynamical systems, we implemented a new **graph-agnostic baseline, MLP-ODE** (suggested by **Reviewer s6hu**). The results (updated Figure 1) show that ignoring topological information leads to a catastrophic increase in error, confirming that this task is non-trivial and requires specific graph-aware architectures like GKAN-ODE.
*   We significantly **deepened our robustness analysis**. We implemented a **polynomial interpolation denoising** strategy following *Rudy et al. (2017)* to demonstrate stability under high noise up to 20dB (requested by **Reviewer K8qq**). Additionally, we performed an **ablation study on derivative estimation**, comparing the 5-point stencil vs. Central Finite Differences (requested by **Reviewers s6hu and K8qq**), confirming that our model rankings are stable and not artifacts of pre-processing.
*   We improved the presentation by adding a comprehensive **framework pipeline figure** in Appendix B.2 (requested by **Reviewer K8qq**) to illustrate the information flow. We also expanded the experimental comparison to include the **original KAN symbolic algorithm** (suggested by **Reviewers s6hu and Usk1**), demonstrating that our proposed Spline-Wise method achieves a superior trade-off between accuracy and complexity.

We have marked all major revisions in the updated manuscript in $\color{blue}\text{blue}$. Additionally, for improved readability and to facilitate the verification of our new results, we have **integrated the Appendix directly at the end of this PDF document**, rather than keeping it as a separate supplementary file.

We believe these new experiments and clarifications provide the evidence requested by the reviewers, and we look forward to a fruitful discussion.

---

### Author Response · Authors · 2025-12-03
**Message to AC and Summary of the Rebuttal**

Dear Area Chair,

In light of the changes to the review process, we would like to provide you with this summary of the reviews and our rebuttals to assist you in your assessment. We have taken the reviewers' constructive feedback seriously and conducted extensive additional experiments, which we believe have substantially strengthened the rigor and quality of our work.

**A more detailed breakdown of our key revisions is reported in the General Statement immediately following this message.**

The reviewers expressed a strong consensus on the quality of our work, commending the manuscript for its **clarity and presentation** (Reviewer Usk1, s6hu). They specifically highlighted the **value of our rigorous benchmarking in addressing a significant gap in the literature** (Reviewer s6hu, jmyg) and **praised the originality of the GKAN-ODE framework for successfully recovering interpretable symbolic laws** (Reviewer Usk1, K8qq, s6hu, jmyg).

Regarding the discussion phase prior to the reversion, we were encouraged by the engagement of **Reviewer s6hu**, who explicitly stated that all their concerns were addressed by our new experiments and subsequently raised their score from 6 to 8. We would have also valued the opportunity to hear back from **Reviewer Usk1**, who was initially "sympathetic to the concept" and "happy to be corrected" regarding their concerns on benchmarks and interpretability, which we believe our rebuttal has addressed. Similarly, **Reviewer K8qq** raised valid points regarding empirical robustness (integrators, noise) rather than the method's core validity; we addressed each of these with new ablations and analyses and hoped to confirm that these satisfied their requirements.

We are confident that our revisions address the core issues raised, and we hope this summary facilitates a smooth re-evaluation process.

Thank you again for your time and effort.

The Authors

---

### Meta-Review · Area_Chair_Y8Nr · 2026-01-07

**Summary:**

The reviewers have raised the following concerns:

W1. The proposed benchmark is not extensive enough (i.e., it contains a limited set of tasks) to serve as a rigorous benchmark.

W2. Several advantages of the design lack explicit empirical support, including moving nonlinearity from nodes to edges, distilling from a non-symbolic model to a symbolic model to improve interpretability, using splines as intermediaries rather than symbolic nonlinearities, and using multiplicative nodes without hyperparameters to enhance physical interaction modeling. The lack of validation for these claims makes the proposed method’s advantages insufficiently convincing.

W3. The writing of the paper does not clearly convey the design rationale of the proposed approach, making it difficult to understand the key motivation and advantages of the method.

W4. Lack of integrator robustness analysis, making the stability and reliability of the results insufficiently validated.

W5. Lack of in-depth analysis of noise robustness.

W6. Lack of an intuitive illustration of the overall architecture.

Reviewer Usk1 and Reviewer K8qq provided negative ratings (2) before the rebuttal and may maintain or increase their ratings to 4 due to the partial addressing of their concerns. Reviewer s6hu explicitly indicated maintaining their rating at 6. Reviewer jmyg indicated unfamiliarity with the field of this paper.

Overall, this paper is borderline, and the writing and experimental validation could be further improved before final acceptance. Therefore, this paper is not recommended for acceptance at this time.

**Reviewer Concerns:**

During the rebuttal, the authors provided further clarifications and discussions, as well as new experimental results and analyses, to address the raised concerns. In particular, the authors provided new experimental results to partially address W2, W4, and W5; clarifications to partially address W3; and a new illustrative figure to address W6.

Some concerns were not fully addressed, including covering more tasks to compose a more rigorous benchmark, demonstrating improved interpretability through distillation (the authors only clarified why the KAN-based approach has greater interpretability, but not why distillation improves interpretability), and providing explicit empirical results to validate alternative design choices.

**Reviewer Scores:**

Reviewer Usk1 and Reviewer K8qq provided negative ratings (2) before the rebuttal and may maintain or increase their ratings to 4 due to the partial addressing of their concerns. Reviewer s6hu explicitly indicated maintaining their rating at 6. Reviewer jmyg indicated unfamiliarity with the field of this paper.

---

### Decision · Program_Chairs · 2026-01-26

Reject